# Assessing global drinking water potential from electricity-free solar water evaporation device

Wei Zhang [1,2,3], Yongzhe Chen [4], Qinghua Ji [1] ✉, Yuying Fan [2,5], Gong Zhang [1], Xi Lu [1], Chengzhi Hu [2,3], Huijuan Liu [1] & Jiuhui Qu [1,2,3] ✉

Universal and equitable access to affordable safely managed drinking water (SMDW) is a significant challenge and is highlighted by the United Nations' Sustainable Development Goals-6.1. However, SMDW coverage by 2030 is estimated to reach only 81% of the global population. Solar water evaporation (SWE) represents one potential method to ensure decentralized water purification, but its potential for addressing the global SMDW challenge remains unclear. We use a condensation-enhanced strategy and develop a physics-guided machine learning model for assessing the global potential of SWE technology to meet SMDW demand for unserved populations without external electricity input. We find that a condensation-enhanced SWE device (1 m²) can supply enough drinking water (2.5 L day⁻¹) to 95.8% of the population lacking SMDW. SWE can help fulfill universal SMDW coverage by 2030 with an annual cost of 10.4 billion U.S. dollars, saving 66.7% of the current investment and fulfilling the SDG-6.1 goal.

Safely managed drinking water (SMDW) is an urgent requirement for people worldwide and has been included in the Sustainable Development Goals (SDG) 6.1 framework[1,2]. For better drinking water accessibility, global populations tend to live close to surface freshwater sources. Even in arid areas that suffer from limited surface water bodies, the median population distance to surface water is 4.3 km for Northern Africa and 4.8 km for the Middle East[3]. Meanwhile, ~ 95% of the population without SMDW live in areas with over 20 cm of annual precipitation (Supplementary Fig. 1). Surface water, groundwater, and rainwater collectively comprise the water source for the population without SMDW[3,4]. However, over 2 billion people still suffer from unsafe drinking water by 2015, which mainly arises from limited water treatment and poor water management. This situation has become particularly acute for populations in remote areas, who are concurrently threatened by unmanaged water sources, poverty, under-developed purification technology, and isolated population distribution. Traditional routes of centralized water treatment to

ensure SMDW are energy- and capital-intensive and rely on the scale advantage to minimize the treatment cost. These features demand large investment and aggravate the inequality of the global water supply[5,6], challenging the fulfillment of goals outlined in the SDG−6.1 framework[7,8]. The United Nations-estimated SMDW coverage by the year 2030 is only 81% of the global population, making the prospect of achieving 100% coverage of SMDW bleak[9]. Point-of-use water treatment techniques capable of providing SMDW are highly desired, especially in remote areas, with minimized upfront capital investment and reduced adverse environmental impact.

Solar water evaporation (SWE) converts solar energy to heat to initiate water evaporation to purify water from different sources to supply SMDW. Salts, heavy metal ions, organics, and pathogenic microorganisms could be removed from the water. The SWE technology is flexible, feasible, cost-and-energy efficient, and has a near-zero carbon footprint, which is believed to satisfy SMDW demand in remote areas[10–12]. Solar evaporators, encompassing 0 Dimension (0D)/

[1]Center for Water and Ecology, State Key Joint Laboratory of Environment Simulation and Pollution Control, School of Environment, Tsinghua University, Beijing, China. [2]Key Laboratory of Drinking Water Science and Technology, Research Center for Eco-Environmental Sciences, Chinese Academy of Sciences, Beijing, China. [3]University of Chinese Academy of Sciences, Beijing, China. [4]Department of Geography, The University of Hong Kong, Hong Kong, China. [5]School of Environment, Northeast Normal University, Changchun, China. ✉e-mail: qhji@tsinghua.edu.cn; jhqu@tsinghua.edu.cn

1D suspended evaporators (e.g., metal nanoparticles) to 2D interfacial evaporation film (e.g., carbon cloth) and then to 3D evaporators with larger surface areas (e.g., umbrella and tree-shaped designs) have been proposed with a solar-to-vapor efficiency of over 90% under natural irradiance (~1 kW m$^{-2}$ h$^{-1}$) across different water body types (sewage, seawater, brackish water, etc.)[13,14].

However, a gap still exists between the evaporated vapor and the collected SMDW due to mismatches between rapid evaporation and weak condensation[15–18]. Important developments, including larger condensing areas, condensing materials with higher thermal conductivity, forced condensing, and multi-stage devices with latent heat recovery or driven by additional photovoltaics, have been proposed to further SMDW output even with solar-to-water efficiency over 100%[15,19–22]. However, the water production cost of SWE is decided by the efficiency, cost and lifetime simultaneously. Advanced solar evaporators and condensing surfaces tend to increase the cost of raw materials. Forced condensation requires additional electricity input, which weakens the SWE inherent merits of low-cost, flexible implementation and hampers its feasibility to the point-of-use water supply, especially in remote areas[23]. Meanwhile, previous reports relied primarily on laboratory-scale testing supplemented by outdoor operations under favorable weather conditions, and comparing the actual SMDW production performance of different SWE devices in natural conditions is necessary for its objective evaluation[24]. A geospatial tool (AWH-Geo) has been proposed to combine the material water yield kinetics of previous reports with dominant environmental variables to assess the global potential for harvesting drinking water from the air given available climatic resources[25]. It pinpoints the maximum impacts of atmospheric water harvesting to address water scarcity on a global scale, proving a great paradigm for evaluating a technique's contribution to the SDGs. Therefore, it is important to reconsider the SWE technique for better supplying SMDW to serve SDG−6.1. Differently, considering that low-cost and flexible implementation make SWE unique, cost and efficiency should be considered to evaluate the feasibility of SWE under natural conditions. An effective tool that could anticipate the technical and economic potential of SWE and, in turn, reveal the technical bottlenecks to guide the SWE device design, is useful.

Here, we propose a physics-guided machine learning model that integrates the physical model and random forest (RF) method. It can simultaneously unveil the mass–energy transfer mechanisms in SWE devices and assess the SMDW yield potential of the SWE technique both technically and economically. The physical model is based on pilot experiments under natural environmental conditions to clarify the principle of designing the SWE devices and establish the causality between SWE devices and meteorological parameters. With this causality, the physical model is abstracted to the RF method to simplify the calculation. The cost evaluation is also included and is merged with the SMDW yields of SWE and the population without SMDW to inspect the feasibility of SWE. The results show that the SWE proves to be suitable for extending SMDW services globally, as the SWE technique can produce enough clean water, particularly in remote areas that are facing SMDW risks. Enhancing the condensation process in SWE devices can greatly promote SMDW production in different regions worldwide. The total cost can be conserved and comprises only 1/3 of the current investment, which is promising for promoting worldwide SMDW service at the household level both geographically and technically, especially in developing areas. This is of great importance for relieving the cost and time pressure of fulfilling SDG-6.1 goals by 2030.

## Results
### Geography of global water scarcity
To assess the potential impact of the SWE technique, the raster gross national income (GNI) data and the people without SMDW service were first mapped together (Fig. 1). The seamless fabric of both national and subnational survey regions describes the global distribution of the GNI per capita. By coincidence, the population without SMDW service was mainly distributed in low- and lower-middle-income countries (classified based on the GNI per capita, see methods for more details, Supplementary Fig. 2). Over 1783 million people without SMDW service are in these countries, which comprise almost 81.2% of the entire global population facing unsafe drinking water supply. Meanwhile, the population density without SMDW service sharply decreases from

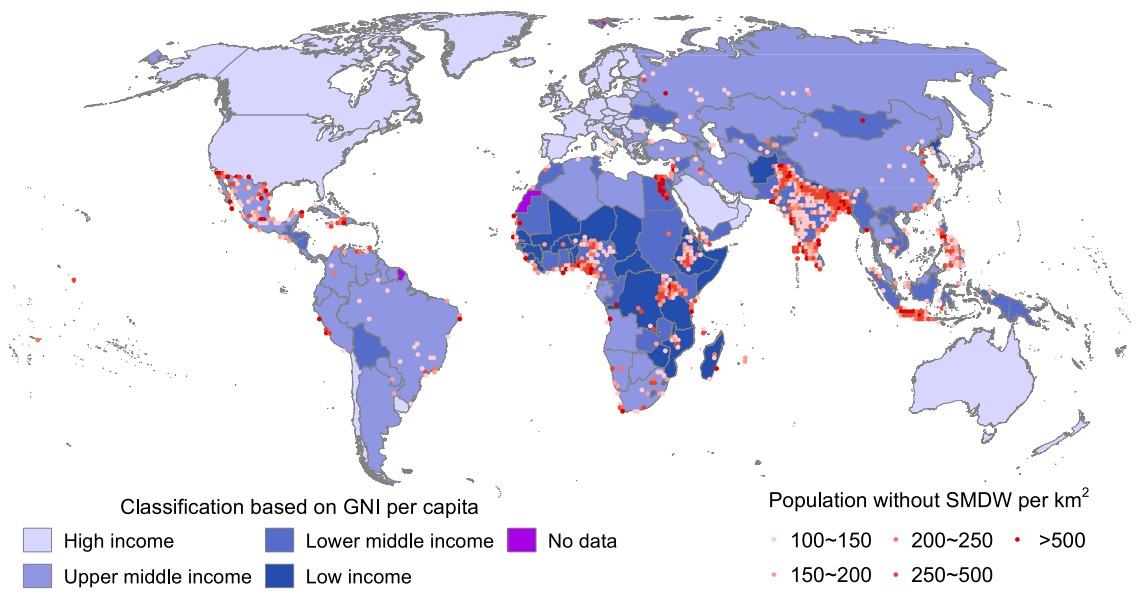

**Classification based on GNI per capita**

High income | Lower middle income | No data
Upper middle income | Low income

**Population without SMDW per km²**

· 100~150 · 200~250 · >500
· 150~200 · 250~500

**Fig. 1 | Geography of the global water-economy nexus.** The World Health Organization/ United Nations International Children's Emergency Fund Joint Monitoring Program (JMP) data reflect the drinking water service levels of each area. Only areas with a population density without safely managed drinking water (SMDW) service >100 km$^{-2}$ are marked. America, Europe, Saudi Arabia, and Australia are high-income countries (according to the Gross National Income per capita, GNI, see methods for more details), while most parts of central Africa, peninsular India, and the Philippines are still in the developing stage or even in the poverty stage.

approximately 66 to 0.26 km$^{-2}$ from high-income countries to lower-middle-income countries. Therefore, economic development serves as a critical stressor contributing to water insecurity and inequity, and economic development inevitably challenges the traditional centralized water supply techniques. The cost of meeting the 2030 SDG-6.1 goals on safe drinking water was estimated to be 60.1–89.0 billion U.S. dollars ($) per year, which is almost 3 times the current investment intensity, which makes the cost unaffordable and unsustainable for most areas facing unsafe drinking water supply[26]. Therefore, developing a cost-effective way to provide SMDW service suitable especially for rural areas is an urgent need[27].

## Successive outdoor solar water evaporation for the SMDW

The SWE technique employs three steps to produce purified water, including solar-thermal conversion (solar to heat), vaporization (heat to vapor), and condensation (vapor to water, Fig. 2a). After decades of efforts, the solar-to-heat efficiency has been elevated to over 90%, and the converted heat could then initiate highly efficient vaporization[13,14]. However, to fulfill the solar-to-water process, condensation is also a critical point that determines the overall SMDW yield[28–31]. Simultaneously evaluating different strategies of condensation and evaporation could help cost-effectively promote global potential estimation of SWE for SMDW services[10,16,32]. We set five cases to differentiate the keys for SWE operation (Supplementary Fig. 3). As shown in Fig. 2b, case 1 is a reference system without solar evaporators and relies on bulk-heating evaporation to produce water. In contrast, solar evaporators are included in case 2. With evaporators, case 3 further pumps vapor out through a condensing tube for forced condensation with additional photovoltaics, while case 4 uses coated glass (condensation-enhanced) to condense the water without external energy input. Case 5 integrates both the condensing tube and condensation-enhanced glass for condensation.

The results show the SWE device could purify water and SMDW yield is affected by both the device setup and the daily meteorological conditions (Fig. 2c and Supplementary Figs. 4, 5). The daytime average inner vapor concentration and temperature of the case 1 SWE device are 102.8 (62.9–137.4) g m$^{-3}$ and 41.8 (30.8–56.0) °C, respectively, higher than those of the natural environment (39.2 g m$^{-3}$ and 34.6 °C, Supplementary Fig. 6). The solar evaporator (case 2) SWE device further elevates the inner vapor concentration and temperature to 146.3 (71.1–211.8) g m$^{-3}$ and 47.9 (31.2–62.1) °C, oversaturating the glass surface more and condensing 50.2% more water compared to the case 1 SWE device. Therefore, converting more sunlight to heat in the closed system is a critical step for creating oversaturation and producing condensing water.

Generally, condensation in the closed SWE device mainly originates from the oversaturation of the water vapor on the condensing glass cooled by the ambient air. The results show that although more solar energy is utilized to power the evaporation, the energy efficiency of the case 2 SWE device ranges from 1.1% to 47.1%, just 50.2% higher than that of the case 1 SWE device (0.0%–43.2%, Fig. 2d). Comparatively, with the condensation-enhanced process, the water yields of the case 3–5 SWE devices increase more dramatically. Interestingly, the case 3–5 SWE devices show lower average inner temperatures and vapor concentrations than the case 2 SWE device, but they show higher SMDW yields. Among them, the case 4 SWE device exhibits a greater efficiency of 13.1–84.0%, which is 99.4% and 199.5% higher than those of case 2 and case 1 SWE devices, respectively. The energy efficiency of case 4 shows no relation (Supplementary Fig. 7a, c and Supplementary Table 1, $p > 0.05$, not significant) to the day within the 100-day successive SMDW production test. Instead, the energy efficiency shows significant positive relations (Supplementary Fig. 7b, c and Supplementary Table 1, $p < 0.001$) to the solar irradiance[33,34], demonstrating that the case 4 device operates stably with almost no deterioration. Moreover, the daytime average inner vapor concentration and

temperature of the case 4 SWE device are only 137.8 (73.1–198.0) g m$^{-3}$ and 47.3 (33.5–60.2) °C, respectively (Supplementary Fig. 6c, d). This is because the coated glass could maintain dropwise condensation, providing self-regenerated condensing sites for the saturated water vapor, and maintain effective latent heat release on the coated glass with the external environment[35,36]. Comparatively, pumping out the vapor through the condensing tube elevates the SMDW yield of cases 3 and 5, but their energy efficiency only ranges from 0.6–14.3% and 1.6–15.2% by taking the solar energy used for the electricity consumption of the vapor pump into consideration, respectively, which is even inferior to the case 2 device without enhanced condensation. The lower inner headspace vapor concentration and temperature of case 3 (132.1 g m$^{-3}$, 46.0 °C) and case 5 (125.6 g m$^{-3}$, 44.2 °C) SWE devices indicate heat and vapor inevitably leak during pumping of the inner air and affect their SMDW yield efficiency. Therefore, high temperature and vapor concentration are necessary for oversaturation but are not the determinant factors for producing SMDW services. Condensation is the bottleneck, and how to improve it dominates the SMDW yield more profoundly.

Therefore, enhanced condensation with the solar evaporator can increase the solar-to-vapor efficiency of the SWE device, and it is even more effective than the application of solar evaporators[23]. This is further demonstrated by performing redundancy analysis (RDA). We set the temperature, absolute humidity, wind speed, and downward shortwave irradiation (DSW) as the explanatory variables and the SMDW daily yields of cases 1–5 as the response variables (Fig. 2e). The results show that wind speed shows little relation to the solar SMDW yields (red arrows), while absolute humidity only exhibits a slightly positive relation with SMDW yields. Compared to absolute humidity, the angles between the temperature and SMDW yields decrease, demonstrating a stronger positive influence of temperature. This is due to that SWE devices could interact with natural conditions by heat exchange, which determines the condensation inside the device and the heat loss from the device to the environment. DSW poses the most dominant influence on the SMDW yields with its correlation coefficient with RDA1 of 0.99. Moreover, through optimizing the condensation, the case 3–5 SWE devices show more strongly positive relations to the DSW compared to the case 1 and 2 SWE devices, corresponding to their higher SMDW yields (Fig. 2c). Amony them, case 4 tops the positive correlation with DSW, which agrees well with its best solar energy utilization efficiency of case 4 (Fig. 2d), proving condensation could make the SWE device better utilize solar energy to produce SMDW.

## Physics-guided machine learning for interpretation and prediction of the SMDW yield

To date, data-driven prediction methods have been adopted to establish the relationships between natural and social parameters, but the data-driven prediction methods showed limited capacities to pinpoint the underlying mechanisms[25,37,38]. The 100-day successive outdoor test provided an essential data set to evaluate the practical solar-to-water efficiency of the SWE device[24]. Based on this data set, finite element physical models were set up based on the outdoor test data to uncover the energy and mass transfer conversion in the gray box of the SWE devices and to establish the relation between critical climate parameters and SMDW yield. Moreover, this physical model could simultaneously train the machine learning model for better prediction. Therefore, we proposed a physics-guided machine learning (PGML) model to integrate both the interpretation and the anticipation of the practical SMDW yields (Supplementary Fig. 8, see "methods")[38,39].

We take the case 2 SWE device as an example of the only evaporation-optimized case (Eva. opt.) and the case 4 SWE device as the evaporation-condensation-optimized case (Eva.-cond. opt.). Hourly downward shortwave irradiance and ambient air temperature are input to construct the working conditions, and the corresponding SMDW yield is applied to optimize the model. As shown in Fig. 3a, a

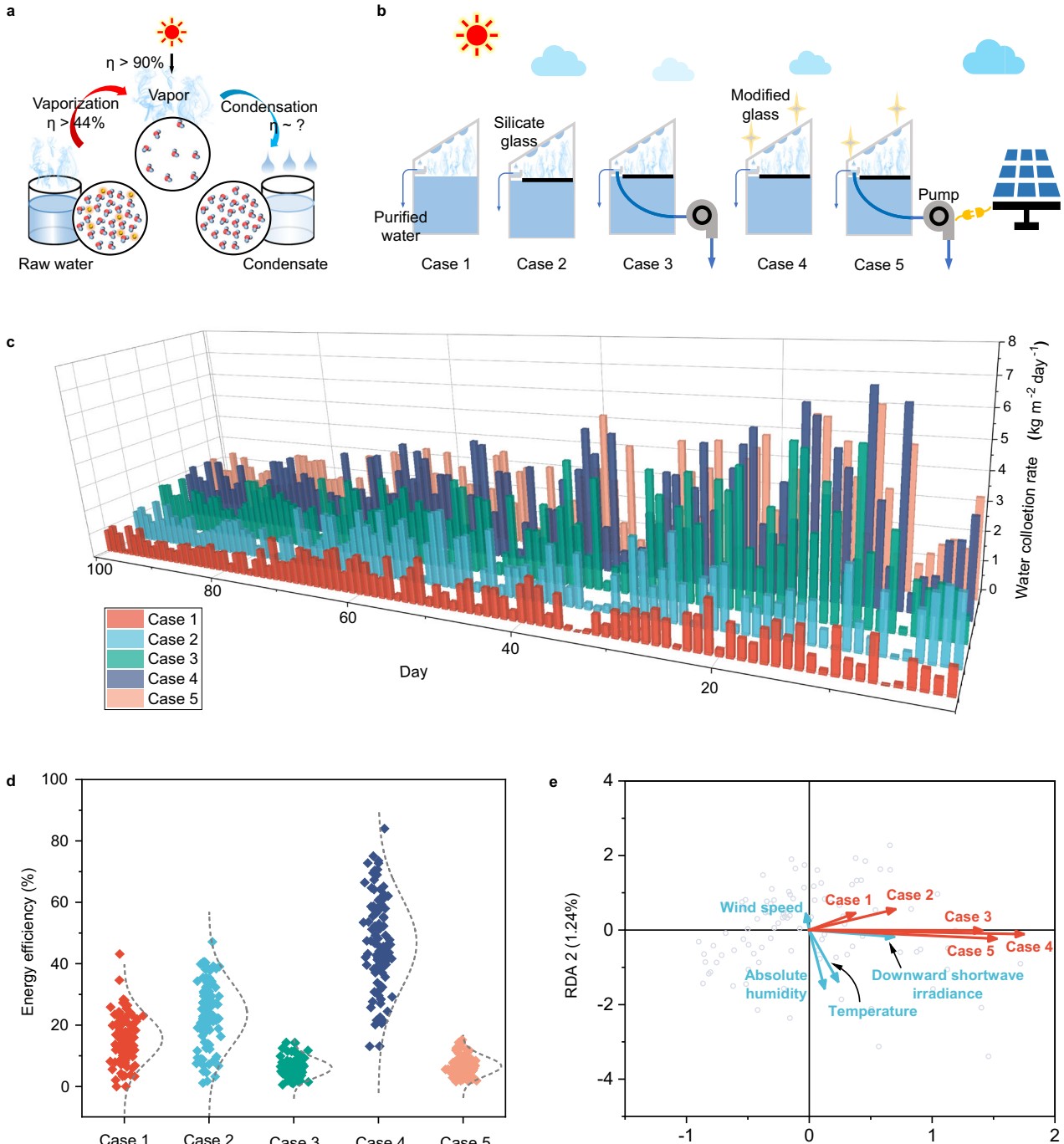

**Fig. 2 | Outdoor evaluation of solar-to-water conversion in solar water evaporation (SWE) devices. a** The solar-to-vapor processes and the corresponding energy efficiency. **b** Schematic diagram of case 1–5 SWE devices. Case 1 is a reference system without solar evaporators. Case 2 includes solar evaporators. Case 3 further pumps vapor out through a condensing tube for forced condensation with additional photovoltaics. Case 4 uses coated glass (condensation-enhanced) to condense the water without external energy input. Case 5 integrates both the condensing tube (powered by photovoltaics) and condensation-enhanced glass for condensation. **c** Daily safely managed drinking water (SMDW) yield of all cases during a 100-day successive pilot study. **d** Statistical distribution of the solar energy utilization efficiency of all cases. **e** Redundancy analysis (RDA) between the meteorological parameters and the SMDW yield.

higher temperature simultaneously accelerates evaporation, enabling a higher vapor concentration and creating oversaturation at the vapor-glass interface. However, even the daytime temperature of the solar evaporator's surface and vapor concentration in the Eva. opt. model is 1.7 °C and 51.0% higher than that of the Eva.-cond. opt. model, but the evaporation rate is 52.1% less and almost halves the corresponding SMDW yield (Fig. 3b). This agrees well with the pilot study results (Supplementary Fig. 6), demonstrating that condensation is the

bottleneck for producing SMDW. Generated vapor is trapped in the closed Eva. opt. model due to limited condensation, which conversely hinders the successive evaporation of the solar evaporator. Therefore, optimizing condensation is the determinate factor of the SWE device, dominating both the solar-to-vapor and the solar-to-water efficiencies.

Moreover, the Eva. opt. the model can sensitively react to meteorological changes, and the simulated SMDW yield is well fitted to the experimental results with a correlation coefficient (*r*) of 0.81

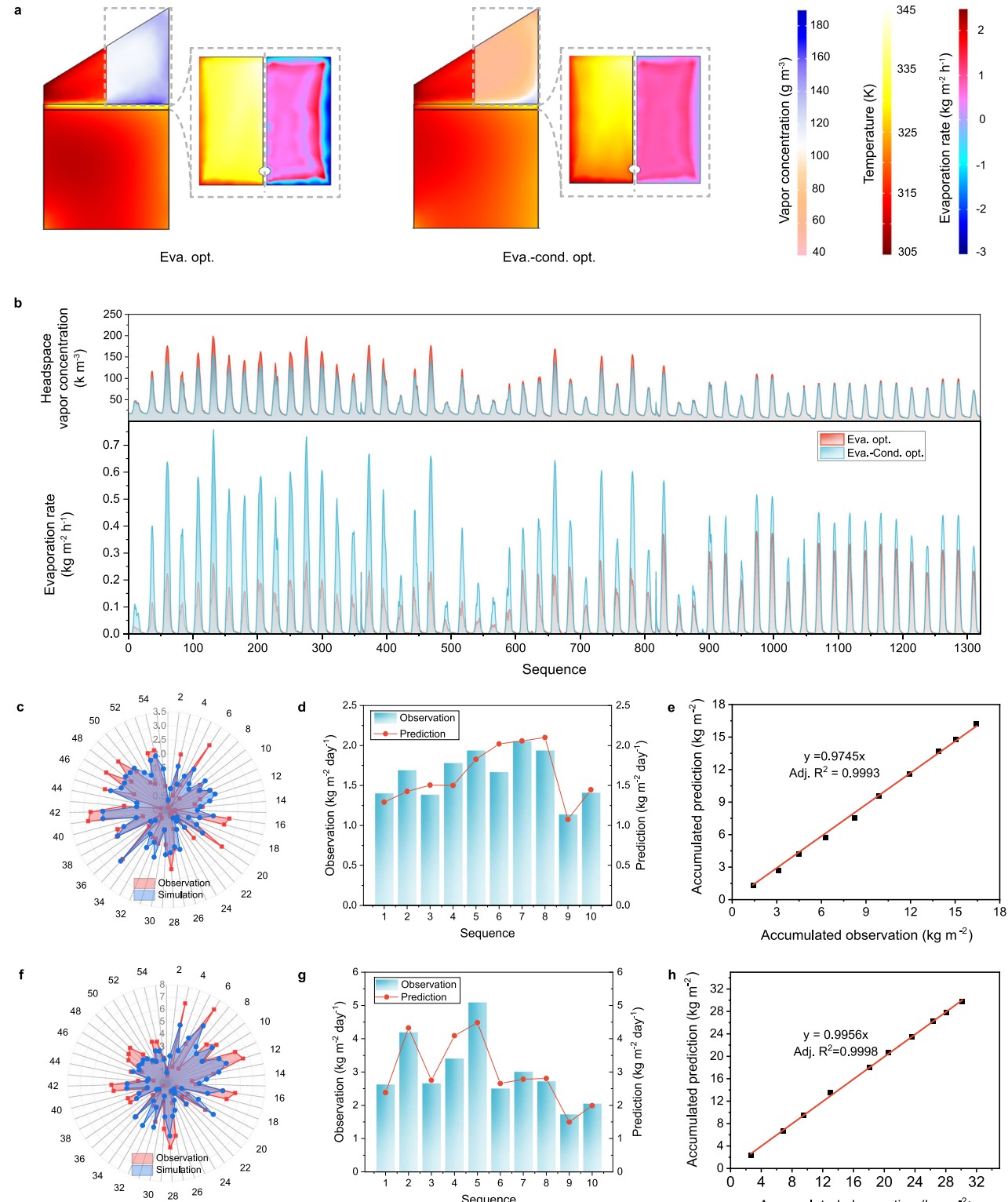

**Fig. 3 | Finite element simulation of the solar water evaporation (SWE) system.**
**a** The inner air temperature, vapor concentration, solar evaporator surface temperature (inset), and evaporation rate (inset) of the SWE devices. **b** Comparisons of the headspace vapor concentration and the solar evaporator surface evaporation rate between the evaporation-optimized case (Eva. opt.) and evaporation-condensation-optimized case (Eva.-cond. opt.) models. **c** Fitting of the Eva. opt. model's simulated safely managed drinking water (SMDW) yield against the observed values. **d** Comparisons between the Eva. opt. model's daily SMDW yield predictions and the observed values. **e** Linear correlation between the accumulated SMDW yield simulation of the Eva. opt. model and the observed value. **f** Fitting of the Eva.-cond. opt. the model simulated the SMDW yield rate against the observed value. **g** Comparisons between the Eva.-cond. opt. model's daily SMDW yield predictions and the observed value. **h** Linear correlation between the accumulated SMDW yield simulation by the Eva.-cond. opt. model and the observed value.

(Fig. 3c). Moreover, when inputting hourly air temperature and surface solar irradiation, the Eva. opt. the model could predict the daily SMDW yield (Fig. 3d), and the correlation coefficient is 0.83. Meanwhile, the accumulated predicted SMDW yield shows a linear correlation with the experimental results with a slope of 0.975 (~1, adjusted $R^2$ is 0.999), demonstrating the robustness of the Eva. opt. model (Fig. 3e). Similarly, the Eva.-cond. opt. the model also well fitted the observed daily SMDW yield with $r$ ~ 0.83 (Fig. 3f). Its prediction of both the daily and accumulated SMDW yields shows an $r$ ~ 0.94 and a slope of 0.996 (adjusted $R^2$ is 0.999), respectively (Fig. 3g, h). Therefore, based on accurately interpreting and comprehensively quantifying the energy and mass transfer processes in the SWE device, the Eva. opt. and Eva.-cond. opt. models establish a causal relationship between the weather parameters and the corresponding SMDW yield and could be generalized for estimating the global SMDW yields of SWE devices.

Based on the SWE finite element physical model, we chose 30 cities that covered the major typical population without SMDW in all continents throughout vast longitudes/latitude ranges and applied the meteorological time series to predict the corresponding SMDW yields (Supplementary Figs. 8, 9). According to the importance of every meteorological input, we trained the tenfold RF models by using both today's and yesterday's DSW (kWh m$^{-2}$) and temperature (K) in these 30 cities as the model predictors and the physical model simulated the daily SMDW yields as the training targets (Supplementary Figs. 8, 10). Then, the potential of the SWE devices to provide the SMDW service globally could be estimated by applying the global meteorological records to the RF models (Supplementary Fig. 10). The RF models showed predicting $R^2$ values of $0.97 \pm 0.0068$ and $0.99 \pm 0.0071$ for the Eva. opt. and Eva.-cond. opt. models, respectively, and the root-mean-square errors (RMSEs) are 0.22 and 0.27 L m$^{-2}$ d$^{-1}$, demonstrating that the RF models have the potential to predict the global SMDW yield.

### Assessing the global SMDW yield potential using a physics-guided machine learning model

To evaluate the technical feasibility of SWE, we first mapped the upper limit of the average annual SMDW yield (L m$^{-2}$ day$^{-1}$) under the hypothesis the DSW could be used for evaporation at 293.15 K, and all generated vapor could be condensed and collected (Fig. 4a)[10,14]. The estimation of this limit is significant for understanding the maximum potential SMDW yield across the world and disclosing the gap between the state-of-the-art and the ideal. As the theoretical yield of SMDW is solely dependent on the abundance of DSW, its geographic pattern closely follows the average annual DSW distribution (Supplementary Fig. 11). The SWE exhibits great potential in globally supplying SMDW services, especially in tropical areas. For areas with large populations without SMDW services, the seasonal variation in the SMDW yield demonstrated the universal feasibility of the SWE in providing local people with almost 10–12 L m$^{-2}$ day$^{-1}$ in the summer (Fig. 4a). In contrast, the SMDW yield in the winter varies dramatically with latitude due to the synchronous reduction in the DSW and the temperature, with only ~2 L m$^{-2}$ day$^{-1}$ for high-latitude cities and over 4 L m$^{-2}$ day$^{-1}$ for tropical and subtropical cities. Therefore, the long-term daily SMDW yield is more suitable for evaluating the availability of the SWE technique than the short-term SMDW yield under favorable conditions.

Then, the annual SMDW yields of two typical electricity-free scenarios (the Eva. opt. and Eva.-cond. opt. models) were mapped. For the Eva. opt. models, even the third quartile of the annual average SMDW yield is only 1.76 L m$^{-2}$ day$^{-1}$, and the SMDW yields in most countries are below this value (Fig. 4b and Supplementary Fig. 12), which fails to satisfy the daily water intake defined by the World Health Organization (2.5 L per capita per day)[40,41]. In contrast, the global median SMDW yield is elevated to 3.77 L m$^{-2}$ day$^{-1}$ for the Eva.-cond. opt. model, with the yield in most countries exceeding this value (Fig. 4c and

Supplementary Fig. 12). The third quartile of the country-level SMDW yields is 4.27 L m$^{-2}$ day$^{-1}$, more than 2-fold that of the Eva. opt. model (1.76 L m$^{-2}$ day$^{-1}$). As excessively low temperatures pose an adverse influence, the SMDW yields of both scenarios exhibit dramatic seasonal variations. The Eva.-cond. opt. model (up to over 7 L m$^{-2}$ day$^{-1}$) outperforms the Eva. opt. model by more than 3 times (Supplementary Fig. 13), especially in summer. The solar-to-vapor efficiency of the Eva.-cond. opt. the model reaches ~60% of the upper limit. However, the SMDW yield in the winter decreases dramatically, and the gap between the Eva.-cond. opt. model and the Eva. opt. model decreases. The SMDW yields of both models are far less than the upper limit, as the upper limit estimation leaves out the adverse influence of low temperature, which contributes greatly to its high average SMDW yield. Therefore, this also informs ongoing efforts that could still lead to better SMDW yield within the thermodynamic limit.

### Extending SMDW service and advancing SDG-6.1 by SWE

Going beyond technical feasibility, the specific cost of implementing SWE to supply SMDW is also crucial for extending SMDW service. As the population distributed without SMDW service in less-developed areas is harder to satisfy and relies on local government effort and global cooperation, the SMDW yield was further classified and analyzed according to the income levels of different countries. The population without SMDW services and SMDW yield were merged with 0.1 million and 2.5 L m$^{-2}$ day$^{-1}$ as the demarcation point of the quadrant (Fig. 5a). For Eva.-cond. opt. model, the high-income countries (classifications are included in the methods–cost evaluation of SWE) are mainly distributed in the second and third quadrants, indicating their low demand for SWE application (Fig. 5a and Supplementary Fig. 14). In comparison, the remaining underdeveloped countries facing SMDW risks almost fall within the first quadrant, exactly matching the SWE resources. Therefore, condensation-enhanced SWE is extremely suitable for underdeveloped areas and it could provide these critical portions of the population lacking SMDW service with over 2.5 L m$^{-2}$ day$^{-1}$ annually, securing individual daily drinking water within 1 m$^2$ of the working area. However, this is impossible for the Eva. opt. the system, as its SMDW supply potential is limited and shows little variance across countries of different income levels (Supplementary Figs. 15, 16). Further reaching the upper limit of SWE, some high-income countries could ascend out of the third and fourth quadrants, but SWE is not urgently needed there (Supplementary Figs. 17, 18). Fig. 5b shows the output service coverage of different systems normalized by the SMDW yield. For the Eva.-cond. opt. model, 95.8% of the population without SMDW is covered by climates favorable for SWE use, where 1 m$^2$ could sustain at least 1 person's daily drinking water. This is superior to the Eva. opt. model, which only covers 2.8% of the total population. When doubling the working area of the Eva-cond. opt. model to 2 m$^2$, its performance could close the gap and even surpass the upper limit of the SWE with a working area of 1 m$^2$. Moreover, by finely recovering the heat of the generated vapor during condensation, a multi-stage design may further elevate the SMDW yield, enabling reinforcement of SMDW services in underdeveloped countries. However, this inevitably leads to extra complexity and cost, which dampens the affordability and accessibility of SWE techniques[42].

To achieve SDG-6.1 goals, the World Bank has estimated the total cost needed to promote SMDW coverage[26]. Ideally, an average annual cost of $80.3 billion (range: 60.1 to $89.0 billion) is needed to extend SMDW services to the unserved population (the cyan dashed line and bars shown in Fig. 5c). However, the actual investment is only 1/3 of the ideal, and this large gap leads to severe delays in the development of further SMDW coverage[26]. Until 2020, the coverage is only 74%, far less than the expected coverage of 81.6% (blue dashed line and bars). Under the current investment trend, the coverage expected by 2030 would be 81%, which fails to meet SDG-6.1 goals[9]. According to the methods of the World Bank, we mapped the capital expense (without

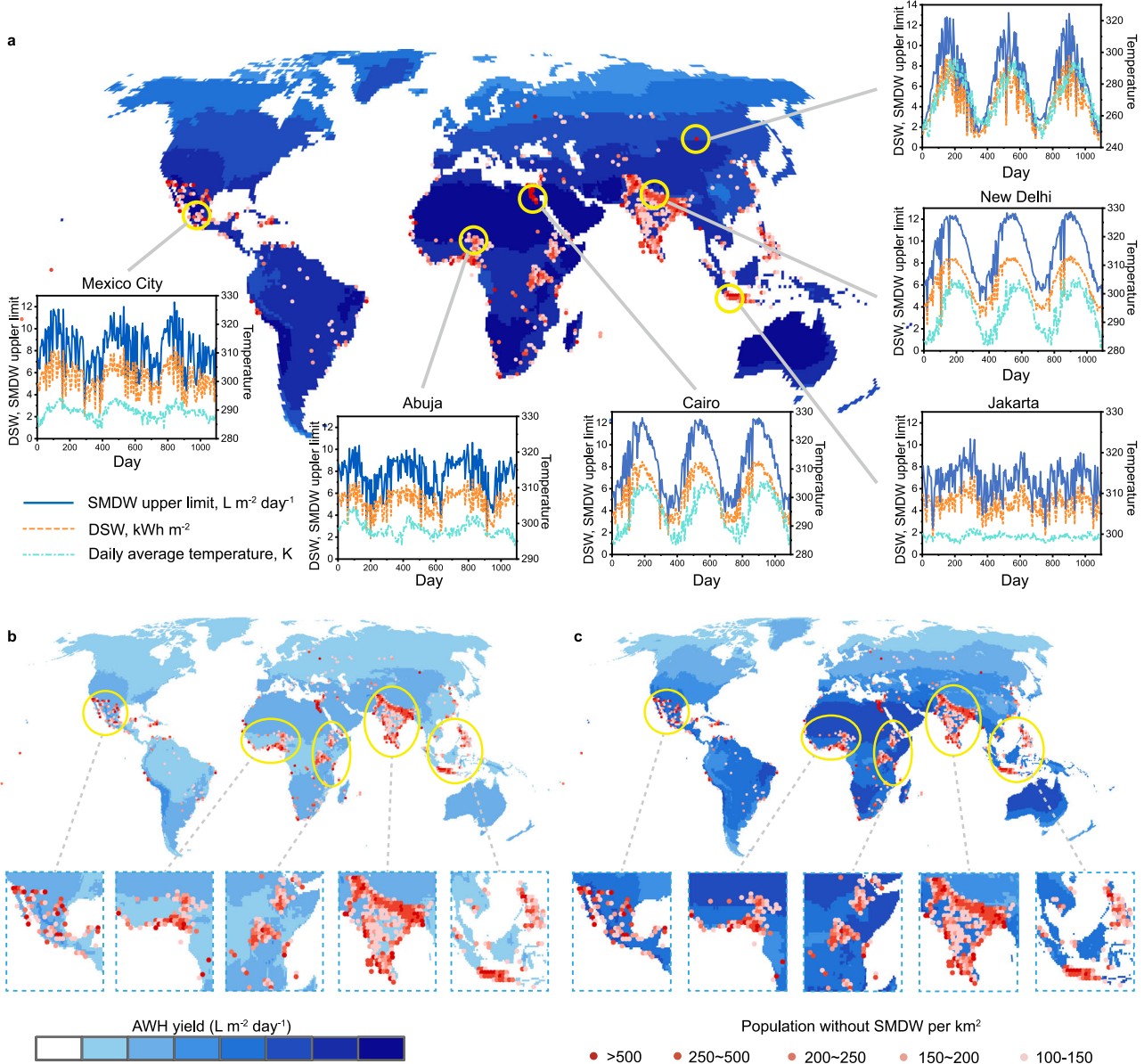

**Fig. 4 | The potential safely managed drinking water (SMDW) yield of solar evaporation and the distribution of the population without the SMDW. a** The upper limit of the mean daily SMDW yield of solar evaporation (the solar-to-vapor efficiency is 100%, without heat recovery). The insets are the seasonal variations in 6 representative cities across the world. The predicted mean daily SMDW yield of (**b**) evaporation-optimized case and (**c**) evaporation-condensation-optimized case.

software) globally (Supplementary Fig. 19). Then, the average annual costs (capital, maintenance, and operation costs) of the SWE approach to close the gap between current and ideal investment (see "methods") are estimated at $26.1 and $10.4 billion for Eva. opt. and Eva.-cond. opt. models, respectively. Benefiting from the cost-effective design of the coated top cover glass for sustainable condensing, no external energy input and auxiliary facilities are needed for the Eva.-cond. opt. model and its cost only comprise 33.3% of the current investment trend ($29.4 billion), while the cost of the Eva. opt. the model shows little advantage (Fig. 5c). Therefore, the SWE technique is promising and constructive in satisfying the remaining SMDW service-lacking population without changing the current investment trend, which offers a reasonable alternative option to address the impossibility of closing the financing gap (~$800 billion), especially under the destructive burden of COVID-19 on the global economy.

## Discussions

As 2030 approaches, SDG-6.1 is increasingly urgent, considering the current trend of SMDW service coverage. Although SWE shows promise in closing the development gap, significant hurdles still impede its application, which requires the collective promotion of technology, economy, and culture across regions. Globally, the variation in the local climate and human landscape inevitably leads to specialized design and installation of SWE devices. Countries facing unbearable SMDW service-lacking populations and who possess abundant SWE resources are recommended as priorities for promoting SWE (plots in the first quadrant of Fig. 5a). The SWE device could be directly installed on the top of flat-roofed buildings, while an auxiliary holder may be necessary for sloping roofed buildings. Additional support may be needed when used in areas with dense forests to ensure the device with enough DSW. Technology development is only one side of the

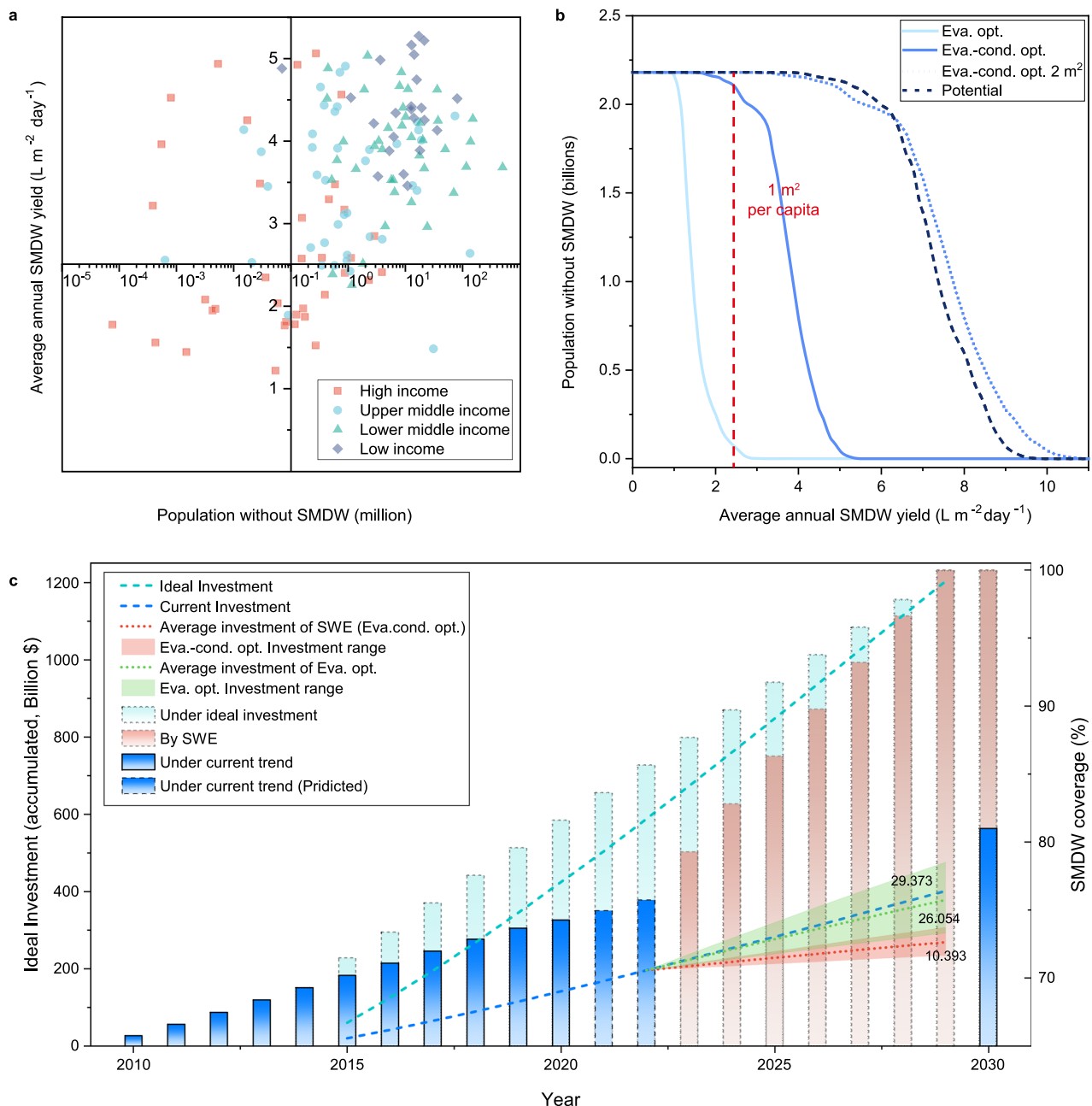

**Fig. 5 | Promoting global safely managed drinking water (SMDW) coverage by solar water evaporation (SWE). a** Four quadrant charts of the annual SMDW yield of the evaporation-condensation-optimized case (Eva.-cond. opt.) concerning the population without SMDW of different income-level countries. **b** The availability of the SMDW produced by evaporation -optimized case (Eva.-opt.) and Eva.-cond. opt.

to the population in need. Doubling Eva.-cond. opt. working area could almost cover the upper potential of the SWE. **c** The relationship between different investment trends and global SMDW coverage (all the bars represent SMDW coverage, and the dashed lines represent investments).

complex route to safe water access, and financing is universally considered critical in promoting SMDW services, but the modes are different considering local economic conditions[43]. For high-income countries, communities could lead the targeted SMDW supply. In upper medium-income countries, governments are suggested to guide and promote SWE in suitable areas. In lower medium- and low-income countries, cross-border cooperation and funds from the United Nations and nongovernmental organizations are essential to compensate for the gap in local funding. Moreover, as the SMDW service-lacking population is mainly distributed in underdeveloped areas, cultural publicity and education on the importance and necessity of SMDW should be complemented to raise local awareness of water security, forming long-term motivation and mechanisms.

In summary, our analysis demonstrates the SWE approach is beneficial for promoting SMDW services technically and economically, especially in areas with the highest needs. Device optimization should be highlighted, especially in cost-effective ways and without complicating the SWE technique. For instance, condensation optimization can greatly promote the SMDW yield throughout the whole year without external energy input. The total cost of promoting SWE to meet SDG-6.1 by 2030 could be 1/3 of the current investment trend, closing the gap with the ideal investment trend (~$800 billion) and even saving on the current cost. This assessment suggests that focusing on SWE device design to make it more effective in electricity-free SMDW production, more flexible in its installation needs, and minimizing cost are worthwhile to assist in meeting the SDG-6.1 stated goals.

## Methods

### Setup of solar evaporation devices

The solar evaporation device was designed with a cuboid-shaped container and a wedge-shaped top cover (Supplementary Fig. 20). The container consisted of an acrylic water tank and a glass top cover. The projection area of the devices is $17 \times 17$ cm². The top of the backboard is 29 cm in height and the inclination of the glass top cover is ~ 22.3°. All device types were installed on the roof of one building in Beijing, China (Supplementary Fig. 3). The whole device was well-sealed and insulated from the ground to avoid direct ground heating. In the devices, the temperature and humidity sensors (TH10S-B, MIAOXIN) were fixed at the top of the backboard, which was connected to an Internet of Things (USR-G781-43, Jinan USR IOT Technology Limited) for data recording. A water storage tank ($35 \times 35 \times 45$ cm) was used to distribute the raw water to the above 5 devices by gravity, and the water level in each container was controlled by a level control valve to 16.0 cm. The raw water was made of artificial brackish water composed of NaCl ($1.0006$ g L⁻¹), CaCl₂ ($0.2775$ g L⁻¹), Na₂SO₄ ($0.4925$ g L⁻¹), and MgCl₂·6H₂O ($0.4177$ g L⁻¹). Diversion trenches were fixed on the four walls of the container to collect the condensed water. The condensed water then got out of the devices through a single silicone tube or a condensing tube immersed in the water, which varied with the 5 device types in our pilot study (Fig. 2b).

Case 1 was a reference system without solar evaporators. Sunlight is directly irradiated into the bulk water to accelerate evaporation and the generated vapor is then condensed on the glass top cover to produce water. Case 2 included solar evaporators to float on the bulk water compared to case 1. The solar evaporators were fabricated via the pyrolysis of the sugarcane, as reported before, and they could utilize ~ 97% of solar energy[44]. Case 3 included solar evaporators and further pumped the headspace vapor through a condensing tube immersed in the bulk water for forced condensation with additional photovoltaics and a vapor pump. The condensed water on the glass top cover and condensing tube comprised the produced water. Case 4 included solar evaporators and used a coated glass top cover (condensation-enhanced) to condense the vapor without external energy input. This coating could ensure the condensed droplets are quickly shed from the glass top cover, so the condensing active sites could be regenerated for enhanced condensation. The fabrication method for the coating layer is included in Supplementary Note 1[36]. Case 5 included solar evaporators and integrated both the condensing tube immersed in the bulk water and the coated glass top cover for condensation. The headspace vapor was pumped through the condensing tube immersed in the bulk water for forced condensation with additional photovoltaics and a vapor pump. The condensed water on the coated glass top cover and condensing tube comprised the produced water.

The daily water output was calculated as Eq. (1):

$$SMDW\ yield\ (\mathrm{kgm^{-2}day^{-1}}) = \frac{mass\ of\ daily\ collected\ water(\mathrm{kg})}{evaporation\ area(\mathrm{m^2})} \quad (1)$$

Where the mass of the daily collected water is measured by a graduated cylinder at 18:00 every day. The conductivity of the produced water from each device was tested by a conductivity meter (S230, METTLER). The ion concentration was determined by anion chromatography (Aquion, Thermo Fisher).

### The SMDW yield and energy efficiency calculations

The solar-to-water energy efficiency was calculated as the ratio between the heat of the generated vapor and the consumed energy[45]

as Eq. (2):

$$\eta_{\mathrm{solar-water}} = \frac{\dot{m} \times h_{\mathrm{fg}}}{q_{\mathrm{solar}} + q_{\mathrm{solar-electricity}}} \quad (2)$$

Where, $\dot{m}$ is mass flow of the SMDW yield (kg m⁻²), $h_{\mathrm{fg}}$ is the latent heat (kJ kg⁻¹), $q_{\mathrm{solar}}$ is the incident solar intensity for heat conversion (kJ m⁻²) and $q_{\mathrm{solar-electricity}}$ is the incident solar intensity for electricity generation (kJ m⁻²). The power of the vapor pump is 5 W, and it works all day to pump the headspace water vapor in a work area of $17 \times 17$ cm². Considering that the photovoltaic cells available in the market have an average energy conversion efficiency of 20%[46], the incident sunlight was estimated as electricity consumption/20%.

### Spatial distribution of population without SMDW service

The income level was determined based on the GNI per capita (https://data.worldbank.org/indicator/NY.GNP.PCAP.PP.CD). The population without SMDW services per km² was derived from Lord et al.'s study[25]. We calculated the population without SMDW per km² at a 1-degree resolution and filtered out the 1° grids with values higher than 100 persons km⁻², which were converted into points afterward.

### Establishing physics-guided machine learning to access the global SMDW yield

Physics-guided machine learning integrates both finite element simulation and the RF model[38,39].

The finite element simulation model was constructed by COMSOL Multiphysics. Modules including single-phase flow (spf), heat transfer (ht), moist transfer (mt) and radiation (rad) are applied to simulate the multi-physical processes in the device (Fig. 3a and Supplementary Fig. 20). For simplification, the 3D model was set with $zOx$ as the symmetrical surface.

In the spf module, turbulent flow (spf1) and laminar flow (spf2) were used to describe the airflow in the device and the outside environment, respectively. The wall is no slip. Gravity was included to induce buoyancy. In spf1, an open boundary was used to ensure the mass and momentum conservation of spf1. The headspace was set as moist air. In spf2, the wind speed of inlet 1 is controlled to 3 m s⁻¹ and the right end of this model was set as an open boundary wet net normal stress of 0 N m⁻².

In the ht module, heat transfer in solids and fluids (ht1) and heat transfer in fluids (ht2) were used to describe the processes in the device and the outside environment, respectively. The equation is as Eq. (3):

$$\rho C_p \left( \frac{\partial T}{\partial t} + \boldsymbol{u} \cdot \nabla T \right) + \nabla \cdot (\boldsymbol{q} + \boldsymbol{q_r}) = \alpha_p T \left( \frac{\partial T}{\partial t} + \boldsymbol{u} \cdot \nabla p \right) + \tau \cdot \nabla \boldsymbol{u} + Q \quad (3)$$

Where $\rho$ is the density, $C_p$ is the specific heat capacity at constant pressure, $T$ is the absolute temperature, $\boldsymbol{u}$ is the velocity vector, $\boldsymbol{q}$ is the heat flux by conduction, $\boldsymbol{q_r}$ is the heat flux by conduction, $\alpha_p$ is the coefficient of thermal expansion, $p$ is the pressure, $\tau$ is the viscous stress tensor, and $Q$ contains heat sources other than viscous dissipation. In ht1, the headspace was set as moist air, and the water domain was set as a solid to simplify the calculation. Meanwhile, a layer with a thickness of 5 mm was set at the upper surface of the water domain (thermal conductivity 0.05 W m⁻¹ K⁻¹) to describe the layer of solar evaporators and the heat localization effect. Its surface temperature originated from the diffuse surface in the rad module. The boundary of ht1 was set as a single layer (thermally thick) to correspond to the glass top cover (1 mm) and acrylic container (4 mm). The bottom of the ht1 is set as a heat-insulating layer to avoid directing heat from the ground. In ht2, the whole domain was set as moist air, and the temperature was derived from the actual hourly temperature (interpolation functions). The inlets were coupled to the inlets in spf2, and

the temperature was also derived from the actual hourly temperature. The bottom of ht2 was set as a heat source coupled to solar irradiance to simulate the effect of the ground.

In the rad module, an out-radiation heat source was applied to simulate the sun. The solar irradiance was obtained by interpolation functions based on the actual hourly shortwave downward irradiance. The light source was set at infinity and the incident direction was controlled to vector (8, 0, −13) for simplicity. The surface of the solar evaporators was set as a diffuse surface.

In the mt module, mt1 and mt2 were used to describe processes in the device and the outside environment, respectively. The equation is as Eq. (4):

$$M_v \frac{\partial c_v}{\partial t} + M_v \boldsymbol{u} \cdot \nabla c_v + \nabla \cdot \left( -M_v D \nabla c_v \right) = G \tag{4}$$

Where $M_v$ is the molar mass of water vapor, $c_v$ is the vapor concentration, $D$ is the vapor diffusion coefficient in air, $\boldsymbol{u}$ is the air velocity field, $G$ is the moisture source (or sink). D was set as a temperature function ($D_v = 0.211(T/273.15)^{1.94} \times (p_0/p)$)[47]. In mt1, a wet surface was applied at the top surface of the solar evaporator to initiate evaporation with an evaporation rate factor of $0.08\,\mathrm{m\,s^{-1}}$. Moist surfaces were applied to the glass top cover to initiate the condensation. The initial liquid water concentration was $0\,\mathrm{mol\,m^{-2}}$ and the evaporation rater factor was optimized according to the 100-day outdoor water yields. The open boundary was coupled to spf1. In mt2, the initial relative humidity of mt2 was 20%.

In the multi-physics module, mt1 was coupled with ht1 and spf1, while mt2 was coupled with ht2 and spf2.

The finite element model was optimized by the obtained SMDW yield of the solar evaporation pilot study. The hourly surface DSW and air temperature were used to define the environmental conditions of the solar evaporation device. The hourly 1° x 1° resolution all-sky surface DSW was obtained from the CER_SYN1deg-1Hour_Terra-Aqua-MODIS_Edition4A dataset, which was retrieved from the Clouds and the Earth's Radiant Energy System (CERES) instruments onboard Terra and Aqua satellites[48]. The hourly 0.5° x 0.625° resolution surface air temperature data comes from the Modern-Era Retrospective analysis for Research and Applications, Version 2, tavg1_2d_flx_Nx dataset, a 2-dimensional assimilated surface flux diagnostics data collection in Modern-Era Retrospective analysis for Research and Applications version 2[49]. The transparency and condensing rate of the glass top cover were optimized to make the SMDW yield of the physical model output match the results of the pilot study. Fifty-five days were selected as the training data set, and 10 random days were chosen as the validation data set to evaluate the performance of the physical model.

As shown in Supplementary Fig. 8, based on the finite element model, an RF model was established as follows. We first randomly selected 30 sites (cities) all around the world, whose latitude and longitude are included in Supplementary Table 2. Secondly, for each city, we randomly selected one 10-day period in the first half of each year between 2019 and 2021, and another 10-day period in the latter half of each year. Thirdly, we utilized the physical model (finite element model), with hourly meteorological data used as inputs, to simulate the hourly SMDW yield of both the Eva. opt. and Eva.-cond. opt. models during each of the six 10-day periods. Fourthly, after averaging the hourly meteorological data and SMDW yield to the daily scale, we compiled the 60 daily surface downward solar shortwave radiation (DSW), surface air temperature, and SMDW at all 30 sites. Fifthly, we developed RF models by using the meteorological data on the day of and the day before SMDW production as the predictors and applying the SMDW yield as the training target. So they can relate the DSWs and temperatures on the day and the day before to the daily SMDW yields.

The obtained RF models correspond to both the Eva. opt. and Eva.-cond. opt. models were checked by 10-fold cross-validation. Meteorological data from the whole world (excluding Antarctica) were used as the RF model inputs to obtain the corresponding SMDW yield map. The daily maps were averaged for 3 years (2019–2021) to obtain the annual average SMDW yield map. Six cities, including Mexico City, Abuja, Cairo, Jakarta, New Delhi, and Ulaanbaatar, were selected to show the daily SMDW yield and the seasonal variation across the three years. Then, the maps of the global SMDW yield and the population without SMDW services were combined to show how the SMDW yield matched the population who needed it. Finally, the SMDW yield across the world was also classified according to the country to obtain the annual average SMDW of different countries.

## Cost evaluation of SWE

According to the World Bank's method of calculating the cost of meeting the 2030 SDG-6.1 targets on drinking water, sanitation, and hygiene, the total cost of SWE was separated into three parts, including the capital, maintenance, and operation costs. The discount rate was set at 5%, and all costs were converted to those existing in 2015 for better comparisons with the ideal and current investment data provided by the World Bank[26]. The capital cost per area is defined as Eq. (5):

$$\begin{aligned} The\ capital\ cost\ per\ area = \ &raw\ materials(\$44.333\mathrm{m^{-2}}) + energy(\$0.265\mathrm{m^{-2}}) \\ &+ facilities(\$1.161\mathrm{m^{-2}}) + labor\ costs(year2022) \end{aligned} \tag{5}$$

The raw materials refer to the solar evaporators, sodium alginate, polyvinyl alcohol (PVA), poly(methyl methacrylate) and glass used. The energy cost refers to the manufacturing energy of devices and fabrication of the solar evaporators. The facility cost refers to the necessary parts to make the solar evaporation device work properly, which mainly includes water containers and silicone tubes. The labor cost refers to the manufacturing fees of the solar evaporation device. The details of the capital cost estimation are included in the supplementary information Note 3 (Supplementary Tables 3 and 4).

The labor cost was estimated by setting China as the typical upper-middle-income country with a labor cost of $20.862\,\mathrm{m^{-2}}$, and the labor costs of other counties were converted by the monthly average earnings (https://ilostat.ilo.org/data/) of different income-level countries as divided by the World Bank in 2022. With the raster data of the annual SMDW yield, the corresponding required SMDW supply area per capita was calculated as Eq. (6):

$$SMDW\ supply\ area = \frac{2.5}{SMDW\ yield} \tag{6}$$

and 2.5 L per capita per day is the daily drinking water of a person as instructed by the WHO. Thus, as Eq. (7):

$$capital\ expense\ per\ capita = SMDW\ supply\ area \times capital\ cost\ per\ area \tag{7}$$

The capital cost per capita of each country can be obtained as Eq. (8):

$$capital\ cost\ per\ capita = capital\ expense\ per\ capita \cdot 1.02 \tag{8}$$

1.02 contains 2% of the matched software costs.

The operation cost refers to the price of the consumable parts in the solar evaporation device that need to be replaced every 2 years. The maintenance cost refers to the repair of the whole solar evaporation device, whose lifespan is estimated as 10 years[50,51]. The maintenance costs were estimated as 3% of the capital costs per year

and comprised 30% of the capital costs (estimation details are included in the supplementary Note 3).

Therefore, the total cost of each country was obtained as Eq. (9):

$$Total\ cost = (capital + maintenance + operation) \cdot population\ to\ be\ served\ with\ SMDW \qquad (9)$$

## Data availability
Data used in this work originated from publicly available sources and are labeled in the context when mentioned. Source data associated with this study are provided. Source data are provided with this paper.

## Code availability
Code is available from the authors upon request.

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

## Acknowledgements

This work was supported by the National Natural Science Foundation of China (No. 52125003 received by C. Hu, 52200101 received by W. Zhang, 52221004 received by H. Liu, and 72025401 received by X. Lu) and the China National Postdoctoral Program for Innovative Talents (No. BX2021149 received by W. Zhang).

## Author contributions

W.Z., Q.J., and J.Q. conceived and designed this research; W.Z., Y.C., and Y.F. performed the research; W.Z., Y.C., Q.J., C.H., and H.L. analyzed the data; W.Z., Y.C., Q.J., G.Z., X.L., H.L., and J.Q. wrote the paper. W.Z., Y.C., Q.J., Y.F., G.Z., X.L., C.H., H.L., and J.Q. discussed, commented on, and revised the manuscript.

## Competing interests

The authors declare no competing interests.
