## [Peer Review File · Nature Communications]

REVIEWER COMMENTS

Reviewer #1 (Remarks to the Author):

The manuscript reports on the condensation-enhanced strategy and develops a physics-guided machine-learning model for assessing the global potential of SWE technology to meet SMDW demand for unserved populations without external electricity input. The obtained results indicated that a condensation-enhanced SWE device (1 m²) could supply enough drinking water (2.5 L day⁻¹) to 95.8% of the population lacking SMDW. However, the novelty of this work should be carefully considered in comparison to previous works Nature (2021) 598, 611–617

Other points should be checked:

Line 105, currently the efficiency is over 100% which is shown in many publications. Please check it out

By using the solar to water evaporation, the concentration of ions decreased. Please carefully describe the reason why?

Other factors such as temperature, humidity, light concentrations, etc. on the efficiency of the device should be discussed.

The calculation of solar to steam efficiency should be discussed as Journal of Power Sources, 2020, 448, 227388

Reviewer #2 (Remarks to the Author):

This study is interesting and fulfills a need in the literature. Atmospheric water harvesting (AWH) has received a good deal of attention, with several studies and global potential estimates in place. However, solar water evaporation (SWE) is a bit less reviewed. This paper provides a global potential and overall starting point for further academic study.

The authors take a similar approach to assessing the global potential as was done in other high-profile studies on water harvesting, but apply it to water evaporation. This is good since it allows comparisons of potentials across methods to better understand the challenges of water-related Sustainable development goals.

The authors have identified condensation as the overlooked "bottleneck" to improve SWE technology. The case is well-made.

The analysis of costs and return on investment to reach more people without SMDW using SWE is compelling.

Here are a few comments/questions/suggestions:

- I'm unclear on the role of the finite element model in relation to the RF model. What time period was the DSW data sampled? Is 55 days at hourly intervals (1320 data points) sufficient for training a machine learning model? Can the authors comment on this?

- Where was the data from the 30 worldwide cities derived? From ground measurements? This is not described.

- The DSW and temperature spatial data from CERES and MERRA 2 should be further described (what spatial resolution, sensor?)

- The units of Fig S9 of DSW should include the spatial normalization (I assume it's kWh/year PER METER?)

- A little more background on SWE is needed: the authors should identify water purification and treatment as the purpose of SWE, and perhaps differentiate it from AWH.

- Line 48: "...encompassing from zero to three dimensions..." needs further explanation.

- Line 142: "...short plank..." is not a phrase I understand

- Line 358: "The solar evaporators were fabricated via the pyrolysis of the sugarcane as reported before and they could" doesn't make sense

- Line 381: The citation of "Jackson et al" is a mistake. It should be "Lord et al".

Reviewer #3 (Remarks to the Author):

This paper investigated the condensation enhanced solar water evaporation (SWE) device and proposed a physics-guided machine learning model to unveil the mass–energy transfer mechanisms in SWE devices and assess the safely managed drinking water (SMDW) yield potential of the SWE technique using global data. The model prediction part is interesting and shows optimistic prospect. However, the discussion on condensation enhanced SWE device seems to be misleading. Besides, there are some important details missing in this paper.

Line 54: "The complexity these strategies weakens the SWE inherent merits of low cost, simple facilities and hampers the deployment to the point-of-use water supply" is one of the important claim to support the development of SWE with enhanced condensation in this work. However, such claim is not rigorous. The water production cost of SWE is decided by the efficiency, cost and lifetime simultaneously. Increasing the efficiency and prolonging the lifetime could also reduce the cost, as long as the cost is well controlled. Meanwhile, the "simple facilities" should not referring to the simple design of SWE device, but the passive operation of SWE device. As long as the device operates without additional electricity input and maintenance, it is a simple system.

Line 105: the author claimed that "condensation has been overlooked". However, this is not true, since many studies discussing the condensation of SWE device like: Desalination, 2008, 230(1-3): 51-61; Solar Energy, 2021, 225: 666-693; Renewable and Sustainable Energy Reviews, 2014, 38: 309-322; Applied Thermal Engineering, 2022, 215: 118941. Although tremendous progress in evaporation enhancement has been witnessed in past years, it does not represent condensation is overlooked, since it has been investigated for a long time.

The water production rates have obvious degradation in the 100-day operation. Will the device have further performance degradation in longer-term operation? Could this support the important "10 years" lifetime of the device?

Although the authors are aware of the importance of "electricity-free" system, two cases shown in Fig.2b still include the use of pump. Even if this could be powered by PV, the moving parts will bring additional maintenance and control systems, which weakens the SWE inherent merits of simple facilities. Please note that the use of vapor pump is quite energy-consuming than liquid pump, and the blade of vapor pump is easy to be broken by droplet.

The test results from the outdoor of the five devices are the basis for physics-guided machine learning. The equipment of the test device should be described appropriately. The calculation method of water yield should be described. The picture of the outdoor test and conditions of the corresponding tests should be provided even in SI.

How did the author evaluate the energy efficiency? Did the author consider the solar input for power generation for case 3 and case 5?

The relationship between physical modeling and machine learning should be presented in a visual form to improve the coherence and readability of the paper.

Fig. 2c should be revised. There are errors in the order of columns representing different cases, which need to be displayed uniformly.

Fig. 4b and Fig. 4c should be revised, which represent the key conclusions of this paper, but the critical information zone is too small to see clearly. It is better to enlarge the key areas to improve clarity.

For the cost analysis of SWE, the detailed calculation process of the raw materials cost should be provided. The data sources of the "energy", "facilities" and "labor costs" used for the capital cost calculation should also be elaborated.

REVIEWER COMMENTS

Reviewer #1 (Remarks to the Author):

The manuscript reports on the condensation-enhanced strategy and develops a physics-guided machine-learning model for assessing the global potential of SWE technology to meet SMDW demand for unserved populations without external electricity input. The obtained results indicated that a condensation-enhanced SWE device (1 m²) could supply enough drinking water (2.5 L day⁻¹) to 95.8% of the population lacking SMDW. However, the novelty of this work should be carefully considered in comparison to previous works Nature (2021) 598, 611–617

Response: Thank you for your suggestions. The work Nature (2021) 598, 611–617 has already been included in our manuscript. Generally, there are three differences in our work:

(1) Different models:

Nature (2021) 598, 611–617 proposed a geospatial tool (AWH-Geo) model that was established based on previously reported results in the lab and theoretical estimation. Our physics-guided machine learning model integrates the physical model and random forest (RF) method and is based on our outdoor experiments under natural conditions. The detailed differences are as follows:

Table The model differences

Different aspects	Nature (2021) 598, 611–617	Our work
Model types	AWH-Geo model	Physical model & random forest (RF) method
Water yield data sources	Logistic regression curve fit to the reported SYs of three data points in the lab (0.21, 3.71 and 9.28 l kWh ⁻¹ at 30, 60 and 90%	Outdoor 100-day solar evaporation water yield tests under natural conditions for 5 different

RH)¹.

cases

Theoretical calculation of the specific water yield according to the Carnot cycle from reference².

Input	Relative humidity, temperature and solar irradiance during the water harvesting period	Daily solar irradiance and temperature on the day of and the day before SMDW production
Output	Specific water yield	Specific purified water and the heat and mass transfer mechanism inside the SWE devices

(2) Different analysis of the technique feasibility:

As pointed out by reviewer 3, the water production cost of SWE is decided by the efficiency, cost and lifetime simultaneously. We simultaneously evaluate the feasibility of SWE from three aspects: the design of SWE devices, the annual average SMDW yield and **the cost of serving 2.5 L SMDW per capita across different designs (Fig. 1 and 5)**.

In Fig.1, we mapped the populations and the Gross National Income (GNI) together to point out that poor SMDW coverage is a technical and economic dual-controlled problem. In Fig. 5 we evaluate the feasibility of SWE both technically and economically by providing the SMDW supply potential and the cost of applying SWE to address SDG–6.1.

In comparison, cost evaluation is not the key point in Nature (2021) 598, 611–617, which mainly assesses the potential of addressing SDGs based on the specific water yield.

Fig. 1 Geography of the global water-economy nexus.

Fig. 5 Promoting global SMDW coverage by SWE.

(3) Different application fields:

Nature (2021) 598, 611–617 is in the field of atmospheric water harvesting and the typical two scenarios are sorption-based materials and cooler–condenser. By contrast, our work fits the area of SWE for water purification. Our target is to figure out how the evaporation and condensation in SWE device design would affect their feasibility in serving SDG–6.1.

Therefore, we have revised the introduction section to highlight the different physics-guided machine-learning model we used and the technical and economic assessment of the SWE in addressing SDGs–6.1:

(Manuscript, Line 66–75) A geospatial tool (AWH-Geo) has been proposed to combine the material water yield kinetics of previous reports with dominant environmental variables to assess the global potential for harvesting drinking water from the air given available climatic resources²⁵. It pinpoints the maximum impacts of atmospheric water harvesting to address water scarcity on a global scale, proving a great paradigm for evaluating a technique’s contribution to the SDGs. Therefore, it is important to reconsider the SWE technique for better supplying SMDW to serve SDG–6.1. Differently, considering that low-cost and flexible implementation make SWE unique, cost and efficiency should be considered to evaluate the feasibility of SWE under natural conditions. An effective tool that could anticipate the technical and economic potential of SWE and, in turn, reveal the technical bottlenecks to guide the SWE device design, is useful.

(Manuscript, Line 76–83) Here, we proposed a physics-guided machine learning model that integrates the physical model and random forest (RF) method. It could simultaneously unveil the mass–energy transfer mechanisms in SWE devices and assess the SMDW yield potential of the SWE technique both technically and economically. The physical model was based on pilot experiments under natural environmental conditions to clarify the principle of designing the SWE devices and establish the causality between SWE devices and meteorological parameters. With this causality, the physical model was abstracted to the RF method to simplify the calculation. The cost evaluation was also included and was merged with the SMDW

yields of SWE and the population without SMDW to inspect the feasibility of SWE.

Also, at the beginning of the section “Assessing the global SMDW yield potential using a physics-guided machine learning model”, we clarified the aim of these sections as:

(Manuscript, Line 248–250) To evaluate the technical feasibility of SWE, we first mapped the upper limit of the average annual SMDW yield (in $L\ m^{-2}\ day^{-1}$) under the hypothesis the DSW could be used for evaporation at 293.15 K, and all generated vapor could be condensed and collected (Fig. 4a).

At the beginning of the section “Extending SMDW service and advancing SDG-6.1 by SWE”, we clarified the main points of this section:

(Manuscript, Line 287–288) Going beyond technical feasibility, the specific cost of implementing SWE to supply SMDW is also crucial for extending SMDW service.

Other points should be checked:

2. Line 105, currently the efficiency is over 100% which is shown in many publications. Please check it out

Response: Thank you for your suggestions. Much progress has been made to elevate the solar-to-vapor efficiency to over 100%. Therefore, we have defined the efficiency as solar-to-heat efficiency to precisely refer to the sunlight being converted to heat process (solar-thermal conversion step), and the sentence has been revised as (Manuscript, Line 116–120) “The SWE technique employs three steps to produce purified water, including solar-thermal conversion (solar to heat), vaporization (heat to vapor) and condensation (vapor to water, Fig. 2a). After decades of efforts, the solar-to-heat efficiency has been elevated to over 90%, and the converted heat could then initiate highly efficient vaporization^{3,4}. However, to fulfill the solar-to-water process, condensation is also a critical point that determines the overall SMDW yield²⁸⁻³¹.”

Also, we have added the description of the solar-to-water efficiency over 100% in the introduction section (Manuscript, Line 55–58) as: “Important developments, including larger condensing areas, condensing materials with higher thermal conductivity, forced condensing and multi-stage devices with latent heat recovery or

driven by additional photovoltaics, have been proposed to further SMDW output even with solar-to-water efficiency over 100%^{15, 19-22}.”

3. By using the solar to water evaporation, the concentration of ions decreased. Please carefully describe the reason why?

Response: Thank you for your suggestions. Sorry for failing to make it clear. The water with decreased ion concentration is the condensate.

We have revised Fig. 2a to avoid misunderstanding of the ion decrease process. As shown in the revised Fig. 2a, during the solar water evaporation (SWE) process, water has undergone an evaporation and condensation cycle. Water molecules in the raw water evaporate to water vapor and get separated from the ions. Then water molecules in the vapor condensed into liquid water (condensate) again with almost no ions in it.

In the manuscript, we have measured the conductivity (Fig. S4) and the main anion concentrations (Fig. S5) of the condensate to evaluate the ion removal performance of SWE.

We have also revised the introduction to better highlight that Solar water evaporation is a technique that could purify raw water:

Solar water evaporation (SWE) converts solar energy to heat to initiate water evaporation to purify water from different sources to supply SMDW. Salts, heavy metal ions organics and pathogenic microorganisms could be removed from the water. The SWE technology is flexible, feasible, cost-and-energy efficient and has a near-zero carbon footprint, which is believed to satisfy SMDW demand in remote areas¹⁰⁻¹². (Manuscript, Line 45–49)

Revised Fig. 2a The solar-to-vapor processes and the corresponding energy efficiency.

Fig. S4 The conductivity of the produced water by SWE.

Fig. S5 The ion concentration of the produced water by SWE. (a) Cl⁻. (b)SO₄²⁻ (The red dashed line is the WHO-defined criteria).

4. Other factors such as temperature, humidity, light concentrations, etc. on the efficiency of the device should be discussed.

Response: Thank you for your suggestions. We have included temperature, humidity, light concentrations and wind speed as explanatory variables to evaluate their

correlation with the efficiency of the device by redundancy analysis (RDA). Fig. 2e has also been updated. The revised parts in the manuscript are as follows:

(Manuscript, Line 166–179) This is further demonstrated by performing redundancy analysis (RDA). We set the temperature, absolute humidity, wind speed and downward shortwave irradiance (DSW) as the explanatory variables and the SMDW daily yields of cases 1–5 as the response variables (Fig. 2e). The results show that wind speed shows little relation to the solar SMDW yields (red arrows), while absolute humidity only exhibits a slightly positive relation with SMDW yields. Compared to absolute humidity, the angles between the temperature and SMDW yields decrease, demonstrating a stronger positive influence of temperature. This is due to that SWE devices could interact with natural conditions by heat exchange, which determines the condensation inside the device and the heat loss from the device to the environment. DSW poses the most dominant influence on the SMDW yields with its correlation coefficient with RDA1 of 0.99. Moreover, through optimizing the condensation, the case 3–5 SWE devices show more strongly positive relations to the DSW compared to the case 1 and 2 SWE devices, corresponding to their higher SMDW yields (Fig. 2c). Among them, case 4 tops the positive correlation with DSW, which agrees well with its best solar energy utilization efficiency of case 4 (Fig. 2d), proving condensation could make the SWE device better utilize solar energy to produce SMDW.

Revised Fig. 2e RDA between the meteorological parameters and the SMDW yield.

5. The calculation of solar to steam efficiency should be discussed as *Journal of Power Sources*, 2020, 448, 227388

Response: Thank you for your suggestions. We have added the reference “*Journal of Power Sources*, 2020, 448, 227388” to help readers for a better understanding of the SMDW yield energy efficiency. The energy efficiency calculation methods are included in the “**Methods**” section of the manuscript as:

(Manuscript, Line 414–421) “**The SMDW yield and energy efficiency calculations.** The solar-to-water energy efficiency was calculated as the ratio between the heat of the generated vapor and the consumed energy:

$$\eta_{\text{solar-water}} = \frac{\dot{m} \times h_{\text{fg}}}{q_{\text{solar}} + q_{\text{electricity}}}$$

Where, \dot{m} is mass flow of the SMDW yield (kg m^{-2}), h_{fg} is the latent heat (kJ kg^{-1}), q_{solar} is the incident solar intensity (kJ m^{-2}) and $q_{\text{electricity}}$ is the energy consumption intensity (kJ m^{-2}). The power of the vapor pump is 5 W and works all day to pump the headspace water vapor in a work area of $17 \times 17 \text{ cm}^2$.”

We also took the suggestions of reviewer 3# (Question 6) to include the solar input for power generation for case 3 and case 5 by calculating the energy consumption of the vapor pump.

As a result, the energy efficiency of the Fig. 2d was updated as follows. We also update the analysis in the manuscript as (Line 154–158) “Comparatively, pumping out the vapor through the condensing tube elevates the SMDW yield of cases 3 and 5, but their energy efficiency only ranges from 1.9–37.1% and 5.7–38.1% by taking the energy consumption of the vapor pump into consideration, respectively, which is even inferior to the case 2 device without enhanced condensation.”

Revised Fig. 2d Statistical distribution of the solar energy utilization efficiency of all cases.

Reviewer #2 (Remarks to the Author):

This study is interesting and fulfills a need in the literature. Atmospheric water harvesting (AWH) has received a good deal of attention, with several studies and global potential estimates in place. However, solar water evaporation (SWE) is a bit less reviewed. This paper provides a global potential and overall starting point for further academic study.

The authors take a similar approach to assessing the global potential as was done in

other high-profile studies on water harvesting, but apply it to water evaporation. This is good since it allows comparisons of potentials across methods to better understand the challenges of water-related Sustainable development goals.

The authors have identified condensation as the overlooked "bottleneck" to improve SWE technology. The case is well-made.

The analysis of costs and return on investment to reach more people without SMDW using SWE is compelling.

Response: Thank you for your kind comments.

Here are a few comments/questions/suggestions:

1. I'm unclear on the role of the finite element model in relation to the RF model. What time period was the DSW data sampled? Is 55 days at hourly intervals (1320 data points) sufficient for training a machine learning model? Can the authors comment on this?

Response: Sorry for the missing information that made you confused. We have revised the description of the physics-guided machine learning in the methods section of the manuscript (Line 485-505) as:

As shown in Fig. S8, based on the finite element model, an RF model was established as follows. We first randomly selected 30 sites (cities) all around the world, whose latitude and longitude are included in Table S2. Secondly, for each city, we randomly selected one 10-day period in the first half of each year between 2019 and 2021, and another 10-day period in the latter half of each year. Thirdly, we utilized the physical model (finite element model), with hourly meteorological data used as inputs, to simulate the hourly SMDW yield of both the Eva. opt. and Eva.-cond. opt. models during each of the six 10-day periods. Fourthly, after averaging the hourly meteorological data and SMDW yield to the daily scale, we compiled the 60 daily surface downward solar shortwave radiation (DSW), surface air temperature and SMDW at all 30 sites. Fifthly, we developed RF models by using the meteorological data on the day of and the day before SMDW production as the predictors and applying the SMDW yield as the training target. So they can relate the DSWs and temperatures

on the day and the day before to the daily SMDW yields.

The obtained RF models corresponding to both the Eva. opt. and Eva.-cond. opt. models were checked by a 10-fold cross-validation. Meteorological data from the whole world (excluding Antarctica) were used as the RF model inputs to obtain the corresponding SMDW yield map. The daily maps were averaged for 3 years (2019–2021) to obtain the annual average SMDW yield map. Six cities, including Mexico City, Abuja, Cairo, Jakarta, New Delhi and Ulaanbaatar, were selected to show the daily SMDW yield and the seasonal variation across the three years. Then, the maps of the global SMDW yield and the population without SMDW services were combined to show how the SMDW yield matched the population who needed it. Finally, the SMDW yield across the world was also classified according to the country to obtain the annual average SMDW of different countries.

Hence, the number of data samples available for RF model training is $(10-1) \times 6 \times 30 = 1620$. Because these data samples contain the meteorological conditions and water yield information across different seasons and different regions, and the number of training samples is generally sufficient for a simple machine learning model (e.g., RF model)^{5,6}, we expect that this RF model can be suitable for daily SMDW yield simulation all year round over the world.

To justify the reliability of this RF model, we performed ten-fold cross-validation. ‘Ten-fold’ means we divided all the training data into ten parts, used 9 of those parts for training while reserving one-tenth for testing each time, and repeated this procedure 10 times to derive ten separate RF models. Ten-fold cross-validation has proven to be a reliable method for machine learning model accuracy assessment⁷. We added in the main text (Line 231-234):

The RF models showed predicting R^2 values of 0.97 ± 0.0068 and 0.99 ± 0.0071 for the Eva. opt. and Eva.-cond. opt. models, respectively, and the root-mean-square errors (RMSEs) are 0.22 and 0.27 $L m^{-2} d^{-1}$, demonstrating that the RF models have the potential to predict the global SMDW yield.

Fig. S8 Data processing workflow of the physics-guided machine learning model. Cylinders indicate the data from the pilot study, CERES and MERRA 2. The rest are table frames (the output documents), rhombus (judgments of the model), rectangles (data processing), and parallelograms (output datasets).

Table S2. Locations of the selected 30 cities over the world

City	Longitude	Latitude
Tijuana	-115.7439	31.7774

Lima	-76.4472	-12.3607
Sao Paulo	-46.5646	-23.5229
Recife	-35.0949	-8.0802
Guatemala	-90.4729	14.5922
Cancun	-86.8254	21.0774
Lagos	3.4156	6.5736
Dakar	-17.3282	14.7607
Cairo	31.3193	30.0944
Addis Ababa	38.7241	9.0008
Zabid	43.3325	14.1623
Kampala	32.5934	0.3286
Blantyre	34.9884	-15.846
Cape Town	18.6595	-33.904
Adelaide	138.7666	-34.9316
Kabul	69.377	34.5693
New Delhi	77.1993	28.6996
Colombo	79.9441	7.1402
Bandung	107.7573	-6.5379
Manila	120.809	14.6857
Urumqi	87.5864	43.8195
Ulaanbaatar	106.8674	48.0789
Saint Petersburg	30.4245	59.9336
Arkhangelsk	40.488	64.5961
Novokuznetsk	87.1117	53.7366
Pyongyang	125.8023	39.0779
Lisbon	-9.1743	38.766
Phoenix	-112.0436	33.4795
Vancouver	-123.1398	49.3626
Paris	2.5218	48.9748

2. Where was the data from the 30 worldwide cities derived? From ground measurements? This is not described.

Response: Thank you for your suggestions. As aforementioned, the description of the physics-guided machine learning in the methods section of the manuscript was revised to include the sources of the data (Line 485-505):

As shown in Fig. S8, based on the finite element model, a RF model was

established as follows. We first randomly selected 30 sites (cities) all around the world, whose latitude and longitude are included in Table S2. Secondly, for each city, we randomly selected one 10-day period in the first half of each year between 2019 and 2021, and another 10-day period in the latter half of each year. Thirdly, we utilized the physical model (finite element model), with hourly meteorological data used as inputs, to simulate the hourly SMDW yield of both the Eva. opt. and Eva.-cond. opt. models during each of the six 10-day periods. Fourthly, after averaging the hourly meteorological data and SMDW yield to the daily scale, we compiled the 60 daily surface downward solar shortwave radiation (DSW), surface air temperature and SMDW at all 30 sites. Fifthly, we developed RF models by using the meteorological data on the day of and the day before SMDW production as the predictors and applying the SMDW yield as the training target. So they can relate the DSWs and temperatures on the day and the day before to the daily SMDW yields.

The obtained RF models corresponding to both the Eva. opt. and Eva.-cond. opt. models were checked by a 10-fold cross-validation. Meteorological data from the whole world (excluding Antarctica) were used as the RF model inputs to obtain the corresponding SMDW yield map. The daily maps were averaged for 3 years (2019–2021) to obtain the annual average SMDW yield map. Six cities, including Mexico City, Abuja, Cairo, Jakarta, New Delhi and Ulaanbaatar, were selected to show the daily SMDW yield and the seasonal variation across the three years. Then, the maps of the global SMDW yield and the population without SMDW services were combined to show how the SMDW yield matched the population who needed it. Finally, the SMDW yield across the world was also classified according to the country to obtain the annual average SMDW of different countries.

Therefore, the data from the 30 worldwide cities was the simulated data of the finite element model, and this physical finite element model was trained and verified by the measured 100-day SMDW yield. The physical finite element model simulates the SMDW yield based on the precise equation to calculate the mass and energy transfer and transformation inside the equipment, which could be demonstrated by the both the SMDW yield and the inner conditions (temperature, humidity, evaporation rate) of the

SWE devices (Fig. 3). In contrast, the random forest (RF) model predicts the SMDW yield based on the mathematical relationship between meteorological conditions and the simulated SMDW yield using the physical model. Meanwhile, the RF model saves computing power and is more suitable for assessing the global SMDW supply potential.

Fig. 3 Finite element simulation of the SWE system. (a) The inner air temperature, vapor concentration, solar evaporator surface temperature (inset) and evaporation rate

(inset) of the SWE devices. (b) Comparisons of the headspace vapor concentration and the solar evaporator surface evaporation rate between the Eva. opt. and Eva.-cond. opt. models. (c) Fitting of the Eva. opt. model's simulated SMDW yield against the observed values. (d) Comparisons between the Eva. opt. model's daily SMDW yield predictions and the observed values. (e) Linear correlation between the accumulated SMDW yield simulation of the Eva. opt. model and the observed value. (f) Fitting of the Eva.-cond. opt. model simulated the SMDW yield rate against the observed value. (g) Comparisons between the Eva.-cond. opt. model's daily SMDW yield predictions and the observed value. (h) Linear correlation between the accumulated SMDW yield simulation by the Eva.-cond. opt. model and the observed value.

3. The DSW and temperature spatial data from CERES and MERRA 2 should be further described (what spatial resolution, sensor?)

Response: Thank you for your suggestions. We have added this information in the methods section (Line 474-480) as “The hourly surface DSW and air temperature were used to define the environmental conditions of the solar evaporation device. The hourly $1^\circ \times 1^\circ$ resolution all-sky surface DSW was obtained from the CER_SYN1deg-1Hour_Terra-Aqua-MODIS_Edition4A dataset, which was retrieved from the Clouds and the Earth's Radiant Energy System (CERES) instruments onboard Terra and Aqua satellites⁴⁵. The hourly $0.5^\circ \times 0.625^\circ$ resolution surface air temperature data comes from the MERRA-2 tavg1_2d_flux_Nx dataset, a 2-dimensional assimilated surface flux diagnostics data collection in Modern-Era Retrospective analysis for Research and Applications version 2⁴⁶.”

4. The units of Fig S9 of DSW should include the spatial normalization (I assume it's kWh/year PER METER?)

Response: Thank you for pointing out our mistake. We have revised the Fig. S11 as follows:

Fig. S11 Global distribution of the surface downward shortwave irradiation (DSW).

5. A little more background on SWE is needed: the authors should identify water purification and treatment as the purpose of SWE, and perhaps differentiate it from AWH.

Response: Thank you for your suggestions. UNEP and we both think that poor water management is the key that leads to limited SMDW availability. Poor water management refers to scientific water source exploitation, effective water treatment techniques and reliable water distribution. Therefore, taking your suggestions, after clarifying the fact that people tend to live near the water sources, we have further clarified water management to water treatment techniques to highlight the treatment purpose of SWE. The revisions are as follows:

(Manuscript Line 33-39) However, over 2 billion people still suffer from unsafe drinking water by 2015, which mainly arises from limited water treatment and poor water management. This situation has become particularly acute for populations in remote areas, who are concurrently threatened by unmanaged water sources, poverty, underdeveloped purification technology, and isolated population distribution. Traditional routes of centralized water treatment to ensure SMDW are energy- and capital-intensive and rely on the scale advantage to minimize the treatment cost.

(Manuscript Line 45-49) Solar water evaporation (SWE) converts solar energy to

heat to initiate water evaporation to purify water from different sources to supply SMDW. Salts, heavy metal ions organics and pathogenic microorganisms could be removed from the water. The SWE technology is flexible, feasible, cost-and-energy efficient and has a near-zero carbon footprint, which is believed to satisfy SMDW demand in remote areas¹⁰⁻¹².

6. Line 48: "...encompassing from zero to three dimensions..." needs further explanation.

Response: Thank you for your suggestions. We explained in the introduction section (Line 49-53) as: "Solar evaporators, encompassing 0D/1D suspended evaporators (e.g. metal nanoparticles) to 2D interfacial evaporation film (e.g. carbon cloth) and then to 3D evaporators with larger surface areas (e.g. umbrella and tree-shaped designs) have been proposed with a solar-to-vapor efficiency of over 90% under natural irradiance ($\sim 1 \text{ kW m}^{-2} \text{ h}^{-1}$) across different water body types (sewage, seawater, brackish water, etc.)^{13,14}."

7. Line 142: "...short plank..." is not a phrase I understand

Response: Thank you for your suggestions. We have replaced the word "short plank" with "bottleneck", and the sentence now has been revised as "Condensation is the bottleneck and how to improve it dominates the SMDW yield more profoundly." (Manuscript Line 162-163)

8. Line 358: "The solar evaporators were fabricated via the pyrolysis of the sugarcane as reported before and they could" doesn't make sense

Response: Thank you for pointing out our mistakes. We have complemented this sentence as "The solar evaporators were fabricated via the pyrolysis of the sugarcane as reported before and they could utilize $\sim 97\%$ of solar energy⁴²." (Manuscript Line 391-393)

9. Line 381: The citation of "Jackson et al" is a mistake. It should be "Lord et al".

Response: Thank you for pointing out our mistakes. We have revised the citation to

“The population without SMDW services per km² was derived from Lord et al.'s study²⁵.” (Manuscript Line 423-424)

Reviewer #3 (Remarks to the Author):

This paper investigated the condensation enhanced solar water evaporation (SWE) device and proposed a physics-guided machine learning model to unveil the mass–energy transfer mechanisms in SWE devices and assess the safely managed drinking water (SMDW) yield potential of the SWE technique using global data. The model prediction part is interesting and shows optimistic prospect. However, the discussion on condensation enhanced SWE device seems to be misleading. Besides, there are some important details missing in this paper.

1. Line 54: "The complexity these strategies weakens the SWE inherent merits of low cost, simple facilities and hampers the deployment to the point-of-use water supply" is one of the important claim to support the development of SWE with enhanced condensation in this work. However, such claim is not rigorous. The water production cost of SWE is decided by the efficiency, cost and lifetime simultaneously. Increasing the efficiency and prolonging the lifetime could also reduce the cost, as long as the cost is well controlled. Meanwhile, the "simple facilities" should not referring to the simple design of SWE device, but the passive operation of SWE device. As long as the device operates without additional electricity input and maintenance, it is a simple system.

Response: Thank you for your suggestions. We strongly agree with your opinion that the water production cost of SWE is decided by the efficiency, cost and lifetime simultaneously, and as long as the device operates without additional electricity input and maintenance, it is a simple system.

We have revised the introduction (Line 55-63) as “Important developments, including larger condensing areas, condensing materials with higher thermal conductivity, forced condensing and multi-stage devices with latent heat recovery or driven by additional photovoltaics, have been proposed to further SMDW output even

with solar-to-water efficiency over 100%^{15, 19-22}. However, the water production cost of SWE is decided by the efficiency, cost and lifetime simultaneously. Advanced solar evaporators and condensing surfaces tend to increase the cost of raw materials. Forced condensation requires additional electricity input, which weakens the SWE inherent merits of low-cost, flexible implementation and hampers its feasibility to the point-of-use water supply, especially in remote areas²³.”

2. Line 105: the author claimed that "condensation has been overlooked". However, this is not true, since many studies discussing the condensation of SWE device like: Desalination, 2008, 230(1-3): 51-61; Solar Energy, 2021, 225: 666-693; Renewable and Sustainable Energy Reviews, 2014, 38: 309-322; Applied Thermal Engineering, 2022, 215: 118941. Although tremendous progress in evaporation enhancement has been witnessed in past years, it does not represent condensation is overlooked, since it has been investigated for a long time.

Response: Thank you for your suggestions. We have replaced this sentence to avoid misleading. The **four references** you recommended have also been included in the manuscript to better support our analysis (Line 117-122).

“After decades of efforts, the solar-to-heat efficiency has been elevated to over 90%, and the converted heat could then initiate highly efficient vaporization^{13, 14}. However, to fulfill the solar-to-water process, condensation is also a critical point that determines the overall SMDW yield²⁸⁻³¹. Simultaneously evaluating different strategies of condensation and evaporation could help cost-effectively promote global potential estimation of SWE for SMDW services^{10, 16, 32}.”

3. The water production rates have obvious degradation in the 100-day operation. Will the device have further performance degradation in longer-term operation? Could this support the important "10 years" lifetime of the device?

Response: Thank you for your suggestions. As shown in Fig. 2c, the SMDW yield seem to decrease versus days, which is due to the solar irradiance are getting weaker during season change. We added an analysis in the manuscript to specify the stable solar

energy conversion efficiency of the case 4 device as (Manuscript Line 146-150):

“The energy efficiency of case 4 shows no relation (Fig. S7a, Table S1, $p=0.44$, not significant) to the day within the 100-day successive SMDW production test. Instead, the energy efficiency shows significant positive relations (Fig. S7b, Table S1, $p=4.7\times 10^{-4}$) to the solar irradiance, demonstrating that the case 4 device operates stably with almost no deterioration.”

Fig. S7, Table S1 and the Spearman coefficients calculation methods have been inserted into the supplementary materials.

Moreover, the proposed device is mainly composed of poly(methyl methacrylate). It is very resistant to UV radiation and other weathering, which possesses a long life with manufacturers' estimated life for many grades 10-20 years even in tropical climates. In some cases poly(methyl methacrylate) could withstand over 20 years of seaside weather involving powerful summer sunshine, the corrosive effects of the salty seaside environment, and winds up to 100mph^{8,9}. Thus, we estimated the life of our solar evaporation device to be 10 years is reasonable, and the maintenance cost has also been included (30% of the capital costs¹⁰) to account for the replacement and fixing of certain parts of the devices.

Fig. S7 The correlation between solar-water energy efficiency and (a) Day, (b) Solar irradiance.

Table S1 The Spearman coefficients between solar-water energy efficiency and Day or Solar irradiance

Spearman coefficients	Day	Solar irradiance
Solar-water energy efficiency	-0.078	0.34
p-value	0.44	4.7×10^{-4} , ***

(Supplementary Information, Line 54-60) Correlation analysis was implemented to test for the effects of day and solar irradiance on the solar-water energy efficiency. The data regarding solar irradiance and solar-water energy efficiency were ln transformed before analysis to minimize the impacts of the outliers. The results show that the solar-water energy efficiency has no significant relation with the pilot experiment day with a p-value of 0.44. On the contrary, the Spearman coefficient is 0.34 between the efficiency and the solar irradiance with a p-value of 4.7×10^{-4} . Therefore, the solar-water energy efficiency of case 4 is stable and no obvious deterioration happens.

4. Although the authors are aware of the importance of "electricity-free" system, two cases shown in Fig.2b still include the use of pump. Even if this could be powered by PV, the moving parts will bring additional maintenance and control systems, which weakens the SWE inherent merits of simple facilities. Please note that the use of vapor pump is quite energy-consuming than liquid pump, and the blade of vapor pump is easy to be broken by droplet.

Response: Thank you for your suggestions. We strongly agree with your opinion that the pump would bring additional maintenance and control systems. In our research, we set five cases to compare different strategies to obtain SMDW. We have made it clear in the manuscript (Line 198-199) that "We take the case 2 SWE device as an example of the only evaporation-optimized case (Eva. opt.) and the case 4 SWE device as the evaporation-condensation-optimized case (Eva.-cond. opt.)." We also reinforced this precondition in the "Assessing the global SMDW yield potential using a physics-guided machine learning model" section (Line 262-263) as "Then, the annual SMDW yields

of two typical electricity-free scenarios (the Eva. opt. and Eva.-cond. opt. models) were mapped.”

By comparison, case 4 tops the SMDW generating efficiency, which only employs solar evaporator and a coated glass top cover (condensation-enhanced) to condense the vapor **without external energy input**.

Then, we build two physics-guided machine learning models to predict the global SMDW supply potential, which corresponds to the case 2 and case 4, respectively. As defined in the manuscript, “Setup of solar evaporation devices” section (**Manuscript Line 388-402**):

Case 1 was a reference system without solar evaporators. Sunlight is directly irradiated into the bulk water to accelerate evaporation and the generated vapor is then condensed on the glass top cover to produce water. Case 2 included solar evaporators to float on the bulk water compared to case 1.

Case 4 includes solar evaporators and used a coated glass top cover (condensation-enhanced) to condense the vapor without external energy input. This coating could ensure the condensed droplets are quickly shed from the glass top cover, so the condensing active sites could be regenerated for enhanced condensation.

Therefore, the two models (Eva. opt. and Eva.-cond. opt. models) do not rely on additional facilities powered by electricity. All the analyses in “Assessing the global SMDW yield potential using a physics-guided machine learning model” and “Extending SMDW service and advancing SDG-6.1 by SWE” sections are free of electricity, corresponding to the simple devices.

5. The test results from the outdoor of the five devices are the basis for physics-guided machine learning. The equipment of the test device should be described appropriately. The calculation method of water yield should be described. The picture of the outdoor test and conditions of the corresponding tests should be provided even in SI.

Response: Thank you for your suggestions. We have added the picture of the outdoor test and conditions as Fig S3. In the main text (**Line 122-123**), we added “**We set five cases to differentiate the keys for SWE operation (Fig. S3)**” for better understanding.

We have also revised the description of the “Setup of the solar evaporation devices” as follows to include the outdoor experiment setups and the calculation method of water yield (Manuscript Line 374-413).

The solar evaporation device was designed with a cuboid-shaped container and a wedge-shaped top cover (Fig. S20). The container consisted of an acrylic water tank and a glass top cover. The projection area of the devices is $17 \times 17 \text{ cm}^2$. The top of the backboard is 29 cm in height and the inclination of the glass top cover is $\sim 22.3^\circ$. All device types were installed on the roof of one building in Beijing, China (Fig. S3). The whole device was well-sealed and insulated from the ground to avoid direct ground heating. In the devices, the temperature and humidity sensors (TH10S-B, MIAOXIN) were fixed at the top of the backboard, which was connected to an Internet of Things (USR-G781-43, Jinan USR IOT Technology Limited) for data recording. A water storage tank ($35 \times 35 \times 45 \text{ cm}$) was used to distribute the raw water to the above 5 devices by gravity and the water level in each container was controlled by a level control valve to 16.0 cm. The raw water was made of artificial brackish water composed of NaCl (1.0006 g L^{-1}), CaCl_2 (0.2775 g L^{-1}), Na_2SO_4 (0.4925 g L^{-1}), and $\text{MgCl}_2 \cdot 6\text{H}_2\text{O}$ (0.4177 g L^{-1}). Diversion trenches were fixed on the four walls of the container to collect the condensed water. The condensed water then got out of the devices through a single silicone tube or a condensing tube immersed in the water, which varied with the 5 device types in our pilot study (Fig. 2b).

- (1) Case 1 was a reference system without solar evaporators. Sunlight is directly irradiated into the bulk water to accelerate evaporation and the generated vapor is then condensed on the glass top cover to produce water.
- (2) Case 2 included solar evaporators to float on the bulk water compared to case 1. The solar evaporators were fabricated via the pyrolysis of the sugarcane as reported before and they could utilize $\sim 97\%$ of solar energy⁴².
- (3) Case 3 included solar evaporators and further pumped the headspace vapor through a condensing tube immersed in the bulk water for forced condensation with additional photovoltaics and vapor pump. The condensed water on the glass top

cover and condensing tube comprised the produced water.

- (4) Case 4 included solar evaporators and used a coated glass top cover (condensation-enhanced) to condense the vapor without external energy input. This coating could ensure the condensed droplets are quickly shed from the glass top cover, so the condensing active sites could be regenerated for enhanced condensation. The fabrication method for the coating layer is included in Supplementary document³⁴.
- (5) Case 5 included solar evaporators and integrated both the condensing tube immersed in the bulk water and the coated glass top cover for condensation. The headspace vapor was pumped through the condensing tube immersed in the bulk water for forced condensation with additional photovoltaics and a vapor pump. The condensed water on the coated glass top cover and condensing tube comprised the produced water.

The daily water output was calculated as:

$$\text{SMDW yield (kg m}^{-2} \text{ day}^{-1}) = \frac{\text{mass of daily collected water (kg)}}{\text{evaporation area (m}^2\text{)}}$$

Where the mass of the daily collected water is measured by a graduated cylinder at 18:00 every day. The conductivity of the produced water from each device was tested by a conductivity meter (S230, METTLER). The ion concentration was determined by anion chromatography (Aquion, Thermo Fisher).

Fig. S3 The picture of the setups of the case 1–5.

6. How did the author evaluate the energy efficiency? Did the author consider the solar input for power generation for case 3 and case 5?

Response: Thank you for your suggestions. We have included the solar input for power generation for case 3 and case 5 by calculating the energy consumption of the vapor pump. The energy efficiency calculation methods are included in the “**Methods**” section in the manuscript as (Line 414-421):

“The SMDW yield and energy efficiency calculations.

The solar-to-water energy efficiency was calculated as the ratio between the heat of the generated vapor and the consumed energy:

$$\eta_{\text{solar-water}} = \frac{\dot{m} \times h_{\text{fg}}}{q_{\text{solar}} + q_{\text{electricity}}}$$

Where, \dot{m} is mass flow of the SMDW yield (kg m^{-2}), h_{fg} is the latent heat (kJ kg^{-1}), q_{solar} is the incident solar intensity (kJ m^{-2}) and $q_{\text{electricity}}$ is the energy consumption intensity (kJ m^{-2}). The power of the vapor pump is 5 W and works all day to pump the headspace water vapor in a work area of $17 \times 17 \text{ cm}^2$.”

As a result, the energy efficiency of the Fig. 2d was updated as follows. We also update the analysis in the manuscript (Line 154-158) as “Comparatively, pumping out the vapor through the condensing tube elevates the SMDW yield of case 3 and 5, but their energy efficiency only ranges from 1.9–37.1% and 5.7–38.1% by taking the energy consumption of the vapor pump into consideration, respectively, which is even inferior to the case 2 device without enhanced condensation.”

Revised Fig. 2d Statistical distribution of the solar energy utilization efficiency of all cases.

7. The relationship between physical modeling and machine learning should be presented in a visual form to improve the coherence and readability of the paper.

Response: Thank you for your suggestions. We have revised Fig. S8 to make the relation of physical modeling and machine learning more intuitive for the reader to understand.

To justify the reliability of this RF model, we performed ten-fold cross-validation. ‘Ten-fold’ means we divided all the training data into ten parts, used 9 of those parts for training while reserving one-tenth for testing each time, and repeated this procedure 10 times to derive ten separate RF models. Ten-fold cross-validation has proven to be a reliable method for machine learning model accuracy assessment⁷. We added in the main text (Line 231-234):

The RF models showed predicting R^2 values of 0.97 ± 0.0068 and 0.99 ± 0.0071 for the Eva. opt. and Eva.-cond. opt. models, respectively, and the root-mean-square errors (RMSEs) are 0.22 and 0.27 $L m^{-2} d^{-1}$, demonstrating that the RF models have the potential to predict the global SMDW yield.

We have also revised the description of the setup of the physics-guided machine learning in the methods section of the manuscript (Line 485-505):

As shown in Fig. S8, based on the finite element model, an RF model was established as follows. We first randomly selected 30 sites (cities) all around the world, whose latitude and longitude are included in Table S2. Secondly, for each city, we randomly selected one 10-day period in the first half of each year between 2019 and 2021, and another 10-day period in the latter half of each year. Thirdly, we utilized the physical model (finite element model), with hourly meteorological data used as inputs, to simulate the hourly SMDW yield of both the Eva. opt. and Eva.-cond. opt. models during each of the six 10-day periods. Fourthly, after averaging the hourly meteorological data and SMDW yield to the daily scale, we compiled the 60 daily surface downward solar shortwave radiation (DSW), surface air temperature and SMDW at all 30 sites. Fifthly, we developed RF models by using the meteorological data on the day of and the day before SMDW production as the predictors and applying the SMDW yield as the training target. So they can relate the DSWs and temperatures on the day and the day before to the daily SMDW yields.

The obtained RF models corresponding to both the Eva. opt. and Eva.-cond. opt. models were checked by a 10-fold cross-validation. Meteorological data from the whole world (excluding Antarctica) were used as the RF model inputs to obtain the corresponding SMDW yield map. The daily maps were averaged for 3 years (2019–2021) to obtain the annual average SMDW yield map. Six cities, including Mexico City, Abuja, Cairo, Jakarta, New Delhi and Ulaanbaatar, were selected to show the daily SMDW yield and the seasonal variation across the three years. Then, the maps of the global SMDW yield and the population without SMDW services were combined to show how the SMDW yield matched the population who needed it. Finally, the SMDW yield across the world was also classified according to the country to obtain the annual average SMDW of different countries.

Fig. S8 Data processing workflow of the physics-guided machine learning model. Cylinders indicate the data from the pilot study, CERES and MERRA 2. The rest are table frames (the output documents), rhombus (judgments of the model), rectangles (data processing), and parallelograms (output datasets).

8. Fig. 2c should be revised. There are errors in the order of columns representing different cases, which need to be displayed uniformly.

Response: Thank you for your suggestions. For better understanding, we have revised

the original Fig. 2c with a new one (Revised Fig. 2c).

We have checked the original Fig. 2c, which is an overlapped (not a stacked) bar plot to show the water collection rate of cases 1-5. As the variation of the daily SMDW yield, the case with the highest SMDW yield would be arranged at the bottom layer and the second highest SMDW yield would be arranged over it and then the third, fourth and fifth highest would be arranged successively. Therefore, the order of the columns could be different.

Now, in the revised Fig. 2c, we have separated cases 1-5 into different y-axis to make the display more uniform.

Revised Fig. 2c Daily SMDW yield of all cases during a 100-day successive pilot study.

Original Fig. 2c Daily SMDW yield of all cases during a 100-day successive pilot study.

9. Fig. 4b and Fig. 4c should be revised, which represent the key conclusions of this paper, but the critical information zone is too small to see clearly. It is better to enlarge

the key areas to improve clarity.

Response: Thank you for your suggestions. We have enlarged the key areas to improve clarity.

Fig. 4 The potential SMDW yield of solar evaporation and the distribution of the population without the SMDW. (a) The upper limit of the mean daily SMDW yield of solar evaporation (the solar-to-vapor efficiency is 100%, without heat recovery). The insets are the seasonal variations in 6 representative cities across the world. The predicted mean daily SMDW yield of (b) Eva. opt. and (c) Eva.-cond. opt. model.

10. For the cost analysis of SWE, the detailed calculation process of the raw materials cost should be provided. The data sources of the "energy", "facilities" and "labor costs" used for the capital cost calculation should also be elaborated.

Response: Thank you for your suggestions. We have added the description of the “raw materials”, "energy", "facilities" and "labor costs" in the method section in the manuscript as (Line 514-519):

The raw materials refer to the solar evaporators, sodium alginate, PVA, poly(methyl methacrylate) and glass used. The energy cost refers to the manufacturing energy of devices and fabrication of the solar evaporators. The facility cost refers to the necessary parts to make the solar evaporation device work properly, which mainly includes water containers and silicone tubes. The labor cost refers to the manufacturing fees of the solar evaporation device. The details of the capital cost estimation are included in the supplementary information (Table S3 and S4).

In the supplementary information we added the calculation processes of these costs as (Line 118-145):

Table S3 The capital cost per area

Classification	Content	Unit price	Usage per area	Cost per area/\$ m ⁻²
Raw materials	Solar evaporator	\$0.070 kg ⁻¹	6.0 kg m ⁻²	0.417
	Sodium alginate	\$3.0 kg ⁻¹	0.0008 kg m ⁻²	0.024
	PVA	\$1.3 kg ⁻¹	0.0008 kg m ⁻²	0.0104
	Poly(methyl methacrylate)	\$2.2 kg ⁻¹	18.17 kg m ⁻²	40.44
	Glass	\$0.24 kg ⁻¹	14.43 kg m ⁻²	3.44
Energy	Electricity	\$0.088 kWh ⁻¹	3 kWh m ⁻²	0.265
Facilities	Water container	-	\$1 m ⁻²	1.0
	Silicone tubes	\$2.55 kg ⁻¹	0.063 kg m ⁻²	0.161
Total				45.7574

The raw material of the solar evaporator is sugarcane, and the cost is its price. The

usage of the sugarcane (kg m^{-2}) is estimated assuming that sugarcane has a density of water (1g cm^{-3}) as $1/(1000\text{g}/1\text{g cm}^{-3}/0.6\text{ cm}) * 10000\text{ cm}^2\text{ m}^{-2}$, where 0.6 cm is the thickness of the solar evaporator (sugarcane).

Sodium alginate (SA), PVA usage was calculated as $200\text{ mL m}^{-2} * 0.04\text{ g}/100\text{ mL}/1000\text{ g kg}^{-1}$, 200 mL m^{-2} is the amount of the cast solution used per square meter and $0.04\text{ g}/100\text{ mL}$ is the concentration of the SA and PVA solutions.

Poly(methyl methacrylate) usage (kg m^{-2}) was calculated considering a device with a floor area of 4 m^2 , container height of 0.2 m, top cover tilting angle of 30° : $(2\text{ m} * 2\text{ m} + 0.2\text{ m} * 2\text{ m} + (0.2\text{ m} + 1.36\text{ m}) * 2\text{ m}/2 * 2 + 1.36\text{ m} * 2) * 0.006\text{ m} * 1.19 * 10^3\text{ kg m}^{-3}/4\text{ m}^2$, where 2 m is the bottom side length, 1.36 m is the backboard height, 0.006 m is the thickness of the poly(methyl methacrylate) plate and $1.18 * 10^3\text{ kg m}^{-3}$ is the density.

The glass usage (kg m^{-2}) is calculated considering a device with a floor area of 4 m^2 , container height of 0.2 m, top cover tilting angle of 30° : $2.32\text{ m} * 2\text{ m} * 0.005\text{ m} * 2500\text{ kg m}^{-3}/4\text{ m}^2$, where 0.005 m is the thickness and 2500 kg m^{-3} is the density of glass.

The cost of electricity is estimated according to the manufacturing energy of devices, and fabrication of the solar evaporators. The cost of facilities mainly includes water containers and silicone tubes.

Table S4 Labor cost estimation

National income levels	Median monthly wage/\$	Labor costs in terms of wages
Low income	106.23	5.68
Lower-middle income	193.41	10.35
Upper-middle income	390.00	20.86
High income	2075.06	111.00

We take the median monthly income of China as a reference and estimate the labor cost to be $\$20.86\text{ m}^{-2}$ according to the actual manufacturing price for the solar evaporation devices. Then the labor cost is categorized into “low-income”, “lower-

middle income”, “upper-middle income” and “high-income” countries (<https://blogs.worldbank.org/search?keyword=country+classification>). The labor costs were obtained by normalizing the median monthly wage (<https://ilostat.ilo.org/data/>) of these four categories of countries.

References in the response letter:

- 1 Zhao, F. *et al.* Super Moisture-Absorbent Gels for All-Weather Atmospheric Water Harvesting. *Adv Mater* **31**, e1806446 (2019).
- 2 Peeters, R., Vanderschaeghe, H., Rongé, J. & Martens, J. A. Energy performance and climate dependency of technologies for fresh water production from atmospheric water vapour. *Environmental Science: Water Research & Technology* **6**, 2016-2034 (2020).
- 3 Zhou, J. *et al.* Development and Evolution of the System Structure for Highly Efficient Solar Steam Generation from Zero to Three Dimensions. *Adv. Funct. Mater.* **29** (2019).
- 4 Liu, Y. *et al.* Advances and challenges of broadband solar absorbers for efficient solar steam generation. *Environ. Sci.-Nano* **9**, 2264-2296 (2022).
- 5 Luan, J., Zhang, C., Xu, B., Xue, Y. & Ren, Y. The predictive performances of random forest models with limited sample size and different species traits. *Fish Res* **227** (2020).
- 6 Millard, K. & Richardson, M. On the Importance of Training Data Sample Selection in Random Forest Image Classification: A Case Study in Peatland Ecosystem Mapping. *Remote Sens-Basel* **7**, 8489-8515 (2015).
- 7 Wong, T.-T. & Yeh, P.-Y. Reliable Accuracy Estimates from k-Fold Cross Validation. *Ieee T Knowl Data En* **32**, 1586-1594 (2020).
- 8 Halliwell, S. M. *Weathering of plastics glazing materials*, Loughborough University of Technology, (1996).

- 9 Gilbert, M. in *Brydson's Plastics Materials (Eighth Edition)* (ed Marianne Gilbert) 75-102 (Butterworth-Heinemann, 2017).
- 10 Hutton & Varughese. The Costs of Meeting the 2030 Sustainable Development Goal Targets on Drinking Water, Sanitation, and Hygiene. (World Bank, 2016).

REVIEWER COMMENTS

Reviewer #1 (Remarks to the Author):

The authors have done their best to improve the quality of this manuscript, which can be published in the present form

Reviewer #2 (Remarks to the Author):

The authors have addressed my comments adequately and sufficiently.

I believe they have also done so for the other reviewer's comments based on my assessment.

I hope this paper gets into publication.

Reviewer #3 (Remarks to the Author):

In this revised manuscript, the authors have eliminated the misleading discussions on condensation enhanced solar water evaporation (SWE) device, and provided the important details related to the outdoor field test and cost analysis. The reviewer appreciates the further discussion on the water production performance of the 100-day operation, although it still needs further discussion.

1. The authors claimed that the solar-to-water energy efficiency is positively related to the solar irradiance in the 100-day operation. However, it is hard to show the relationship based on the scattered data points in Fig. S7. More discussion about the correlation between the energy efficiency and solar irradiance is encouraged.

2. The authors estimated a lifetime of 10 years for the SWE device according to the performance of poly (methyl methacrylate). However, it should be noted that an important but easy-to-fail component of the device could be the coating for condensation, which accounts for the performance enhancement of the SWE device in this study. How about the durability of this condensation coating? Some more discussions and reference papers are encouraged here to further support the estimation of 10-year lifetime.

3. The authors calculated the solar-to-water energy efficiency by taking both the solar energy and electricity consumption into consideration. This can be unreasonable since they have different energy grades. It is recommended to calculate an overall solar energy efficiency by assuming a photovoltaic efficiency to convert the electricity consumption into solar energy consumption.

Responses to the reviewers:

Reviewer #1 (Remarks to the Author):

The authors have done their best to improve the quality of this manuscript, which can be published in the present form

Response: Thank you for your constructive suggestions and solid contributions to our manuscript.

Reviewer #2 (Remarks to the Author):

The authors have addressed my comments adequately and sufficiently.

I believe they have also done so for the other reviewer's comments based on my assessment.

I hope this paper gets into publication.

Response: Thank you for your constructive suggestions and solid contributions to our manuscript.

Reviewer #3 (Remarks to the Author):

In this revised manuscript, the authors have eliminated the misleading discussions on condensation enhanced solar water evaporation (SWE) device, and provided the important details related to the outdoor field test and cost analysis. The reviewer appreciates the further discussion on the water production performance of the 100-day operation, although it still needs further discussion.

1. The authors claimed that the solar-to-water energy efficiency is positively related to the solar irradiance in the 100-day operation. However, it is hard to show the

relationship based on the scattered data points in Fig. S7. More discussion about the correlation between the energy efficiency and solar irradiance is encouraged.

Response: Thank you for your suggestions. We have added two references in the manuscript, which have also reported the positive relations between the solar evaporation efficiency and the solar irradiance:

“The energy efficiency of case 4 shows no relation (Fig. S7a, c, Table S1, $p > 0.05$, not significant) to the day within the 100-day successive SMDW production test. Instead, the energy efficiency shows significant positive relations (Fig. S7b, c, Table S1, $p < 0.001$) to the solar irradiance^{33, 34}, demonstrating that the case 4 device operates stably with almost no deterioration.” **(Manuscript line 146-150)**

References:

33 Yang, J. et al. Functionalized Graphene Enables Highly Efficient Solar Thermal Steam Generation. *ACS Nano* 11, 5510-5518 (2017).

34 Zhang, P., Li, J., Lv, L., Zhao, Y. & Qu, L. Vertically Aligned Graphene Sheets Membrane for Highly Efficient Solar Thermal Generation of Clean Water. *ACS Nano* 11, 5087-5093 (2017).

In addition, we have supplemented the SI **(Supplementary document line 104-123)**

“By linear regression of energy efficiency and the day (Fig. S7a), the p -value of the slope is 0.51, demonstrating that the linear relationship between them is not significant. In contrast, the p -value of the slope between the energy efficiency and the solar irradiance is 0.001, demonstrating a significant linear relationship between them (Fig. S7b).

Moreover, between-group comparisons (Day, Energy efficiency, and Solar irradiance) were performed using independent samples Kruskal-Wallis one-way analysis of variance (ANOVA). The results showed that the correlation between the energy efficiency and the day is not significant ($p > 0.9999$) while the correlation

between the energy efficiency and the solar irradiance is significant ($p < 0.0001$, Fig. S7c).

Correlation analysis was further implemented to test for the effects of day and solar irradiance on solar-water energy efficiency. The data regarding solar irradiance and solar-water energy efficiency were ln transformed before analysis to minimize the impacts of the outliers. The results show that the solar-water energy efficiency has no significant relation with the pilot experiment day with $p > 0.05$. On the contrary, the Spearman coefficient is 0.34 between the efficiency and the solar irradiance with $p < 0.001$.

From the analysis above, it is concluded that solar-to-water energy efficiency is positively related to the solar irradiance while it stays stable within the 100-day operation.”

Fig. S7 The linear regression between solar-water energy efficiency and (a) Day (Slope=-0.03, $p=0.51$, ns), (b) Solar irradiance (Slope=3.6, $p=0.001$, ***). (c) Between-group comparisons of the day, energy efficiency, and solar irradiance (Day vs. Energy efficiency, $p>0.05$, ns; Energy efficiency vs. Solar irradiance, $p<0.0001$, ****).

Table S1 The Spearman coefficients between solar-water energy efficiency and Day or Solar irradiance

Spearman coefficients	Day	Solar irradiance
Solar-water energy efficiency	-0.078	0.34
p-value	0.44	4.7×10^{-4}

2. The authors estimated a lifetime of 10 years for the SWE device according to the performance of poly (methyl methacrylate). However, it should be noted that an important but easy-to-fail component of the device could be the coating for condensation, which accounts for the performance enhancement of the SWE device in this study. How about the durability of this condensation coating? Some more discussions and reference papers are encouraged here to further support the estimation of 10-year lifetime.

Response: Thank you for your suggestions. We feel sorry for failing to make it clear that the condensation coatings, solar evaporators are considered to be replaced every 2 years. Thus, we have revised the description in the section “Cost evaluation of SWE”:

“The operation cost refers to the price of the consumable parts in the solar evaporation device that need to be replaced every 2 years. The maintenance cost refers to the repair of the whole solar evaporation device, whose lifespan is estimated as 10 years^{50,51}. The maintenance costs were estimated as 3% of the capital costs per year and comprised 30% of the capital costs (estimation details are included in the supplementary document, section “Cost estimation”).” (Manuscript line 534-538)

We have added the detailed operation and maintenance estimation in the supplementary information (**Supplementary document line 213-261**)

“Operation and maintenance cost:

Our devices mainly comprise acrylic containers, carbon solar evaporators, top cover glass, hydrogel-based condensation coatings, and connecting silicone tubes.

The **operation cost** refers to the total price of the parts in the solar evaporation device that need to be replaced after a certain period. It mainly includes solar evaporators and hydrogel-based condensation coatings.

The solar absorber used in our study is inorganic biochar, which is stable in the environment. According to previous reports, solar stills composed of typical inorganic carbon-based black paint exhibit a long lifespan ranging from 2-10 years¹⁻³. In addition, during our 100-day outdoor test, the solar absorbers showed no observable deterioration. Thus, its lifespan is estimated at 2 years.

The coating for accelerated condensation is composed mainly of PVA fibers. PVA is chemically stable and widely used in coatings and fibers. As previously reported, PVA coatings are stable (remain hydrophilic and antifogging) after 2-7 months of exposure to a hot and humid environment or daily use^{4, 5}. Moreover, PVA has a strong bond with the matrix, exhibiting resistance to an alkaline environment. They also estimated that the tensile strength of PVA fiber could be preserved after even 60 years of ultraviolet irradiance in a hot environment⁶. In addition, the coating used in our outdoor test shows no obvious deterioration after 100 days, so its lifespan is also estimated at 2 years for simplification of calculations.

Therefore, from Table S3, the solar evaporator, sodium alginate, PVA, and the electricity used for fabrication is a total of \$0.72 m⁻², which is considered as consumable material and should be replaced. The operation cost was comprised of the substitution of the solar evaporators and condensation coatings and the auxiliary software cost (2% of the materials cost).

The **maintenance cost** refers to the repair of the whole solar evaporation device (whole capital cost). For the whole devices mainly made of Poly(methyl methacrylate) (PMMA), its lifetime is estimated as 10 years considering the property of the PMMA^{7, 8}. Therefore, besides the operation cost to replace the solar evaporators and the condensation coatings, the maintenance costs were estimated at 3% of the capital costs per year and comprised 30% of the capital costs to fix the problems that the whole solar evaporation device may encounter.”

Added references:

- 1 Ni, G. *et al.* A salt-rejecting floating solar still for low-cost desalination. *Energ. Environ. Sci.* **11**, 1510-1519 (2018).
- 2 Kabeel, A. E. *et al.* Effect of water depth on a novel absorber plate of pyramid solar still coated with TiO₂ nano black paint. *J Clean Prod* **213**, 185-191 (2019).
- 3 Abdullah, A. S., Younes, M. M., Omara, Z. M. & Essa, F. A. New design of trays solar still with enhanced evaporation methods – Comprehensive study. *Sol Energy* **203**, 164-174 (2020).
- 4 Yang, M. *et al.* Structure and properties of polyvinyl alcohol (PVA)/Al₂O₃ antifogging coating with self-healing performance. *J Coat Technol Res* (2024).
- 5 Yu, X. *et al.* Highly durable antifogging coatings resistant to long-term airborne pollution and intensive UV irradiation. *Mater Design* **194** (2020).
- 6 Silva, F. A., Peled, A., Zukowski, B. & Toledo Filho, R. D. in *A Framework for Durability Design with Strain-Hardening Cement-Based Composites (SHCC): State-of-the-Art Report of the RILEM Technical Committee 240-FDS* (eds Gideon P. A. G. van Zijl & Volker Slowik) 59-78 (Springer Netherlands, 2017).
- 7 Halliwell, S. M. *Weathering of plastics glazing materials*, Loughborough University of Technology, (1996).
- 8 Gilbert, M. in *Brydson's Plastics Materials (Eighth Edition)* (ed Marianne

Gilbert) 75-102 (Butterworth-Heinemann, 2017).

3. The authors calculated the solar-to-water energy efficiency by taking both the solar energy and electricity consumption into consideration. This can be unreasonable since they have different energy grades. It is recommended to calculate an overall solar energy efficiency by assuming a photovoltaic efficiency to convert the electricity consumption into solar energy consumption.

Response: Thank you for your suggestions. We agree that solar heat and solar electricity have different energy grades. Thus, we have revised the energy efficiency calculation method (**Manuscript lines 415-423**)

“The solar-to-water energy efficiency was calculated as the ratio between the heat of the generated vapor and the consumed energy⁴⁵:

$$\eta_{\text{solar-water}} = \frac{\dot{m} \times h_{\text{fg}}}{q_{\text{solar}} + q_{\text{solar-electricity}}}$$

Where, \dot{m} is mass flow of the SMDW yield (kg m^{-2}), h_{fg} is the latent heat (kJ kg^{-1}), q_{solar} is the incident solar intensity for heat conversion (kJ m^{-2}) and $q_{\text{solar-electricity}}$ is the incident solar intensity for electricity generation (kJ m^{-2}). The power of the vapor pump is 5 W and works all day to pump the headspace water vapor in a work area of $17 \times 17 \text{ cm}^2$. Considering that the photovoltaic cells available in the market have an average energy conversion efficiency of 20%⁴⁶, the incident sunlight was estimated as electricity consumption/20%.”

Correspondingly, Fig. 2d has been revised and the analysis are updated as follows: “Comparatively, pumping out the vapor through the condensing tube elevates the SMDW yield of cases 3 and 5, but their energy efficiency only ranges from 0.6 – 14.3% and 1.6 – 15.2% by taking the solar energy used for the electricity consumption of the vapor pump into consideration, respectively, which is even inferior to the case 2 device

without enhanced condensation.” (Manuscript lines 154-158)

Revised Fig. 2d Statistical distribution of the solar energy utilization efficiency of all cases.

Added reference:

46. Morales Pedraza, J. in *Non-Conventional Energy in North America* (ed Jorge Morales Pedraza) 137-174 (Elsevier, 2022).

REVIEWERS' COMMENTS

Reviewer #3 (Remarks to the Author):

All my comments have been well addressed. I recommend its publication in current form.

Reviewer #3 (Remarks to the Author):

All my comments have been well addressed. I recommend its publication in current form.

Response: Thank you for your efforts and suggestions.